# Sparsistency for Inverse Optimal Transport

**Francisco Andrade & Gabriel Peyré**
ENS Paris
{francisco.andrade,gabriel.peyre}@ens.fr

**Clarice Poon**
University of Warwick
clarice.poon@warwick.ac.uk

## Abstract

Optimal Transport is a useful metric to compare probability distributions and to compute a pairing given a ground cost. Its entropic regularization variant (eOT) is crucial to have fast algorithms and reflect fuzzy/noisy matchings. This work focuses on Inverse Optimal Transport (iOT), the problem of inferring the ground cost from samples drawn from a coupling that solves an eOT problem. It is a relevant problem that can be used to infer unobserved/missing links, and to obtain meaningful information about the structure of the ground cost yielding the pairing. On one side, iOT benefits from convexity, but on the other side, being ill-posed, it requires regularization to handle the sampling noise. This work presents an in-depth theoretical study of the $\ell_1$ regularization to model for instance Euclidean costs with sparse interactions between features. Specifically, we derive a sufficient condition for the robust recovery of the sparsity of the ground cost that can be seen as a far reaching generalization of the Lasso's celebrated "Irrepresentability Condition". To provide additional insight into this condition, we work out in detail the Gaussian case. We show that as the entropic penalty varies, the iOT problem interpolates between a graphical Lasso and a classical Lasso, thereby establishing a connection between iOT and graph estimation, an important problem in ML.

## 1 Introduction

Optimal transport has emerged as a key theoretical and numerical ingredient in machine learning for performing learning over probability distributions. It enables the comparison of probability distributions in a "geometrically faithful" manner by lifting a ground cost (or "metric" in a loose sense) between pairs of points to a distance between probability distributions, metrizing the convergence in law. However, the success of this OT approach to ML is inherently tied to the hypothesis that the ground cost is adapted to the problem under study. This necessitates the exploration of ground metric learning. However, it is exceptionally challenging due to its a priori highly non-convex nature when framed as an optimization problem, thereby inheriting complications in its mathematical analysis. As we illustrate in this theoretical article, these problems become tractable – numerically and theoretically – if one assumes access to samples from the OT coupling (i.e., having access to some partial matching driven by the ground cost). Admittedly, this is a restrictive setup, but it arises in practice (refer to subsequent sections for illustrative applications) and can also be construed as a step in a more sophisticated learning pipeline. The purpose of this paper is to propose some theoretical understanding of the possibility of stably learning a ground cost from partial matching observations.

### 1.1 Previous Works

**Entropic Optimal Transport.** OT has been instrumental in defining and studying various procedures at the core of many ML pipelines, such as bag-of-features matching Rubner et al. (2000), distances in NLP Kusner et al. (2015), generative modeling Arjovsky et al. (2017), flow evolution for sampling De Bortoli et al. (2021), and even single-cell trajectory inference Schiebinger et al. (2019). We refer to the monographs Santambrogio (2015) for detailed accounts on the theory of OT, and Peyré et al. (2019) for its computational aspects. Of primary importance to our work, entropic regularization of OT is the workhorse of many ML applications. It enables a fast and highly parallelizable estimation of the OT coupling using the Sinkhorn algorithm Sinkhorn (1964).

More importantly, it defines a smooth distance that incorporates the understanding that matching procedures should be modeled as a noisy process (i.e., should not be assumed to be 1:1). These advantages were first introduced in ML by the seminal paper of Cuturi Cuturi (2013), and this approach finds its roots in Schrödinger's work in statistical physics Léonard (2012). The role of noise in matching (with applications in economics) and its relation to entropic OT were advanced in a series of papers by Galichon and collaborators Galichon & Salanié (2010); Dupuy & Galichon (2014); Galichon & Salanié (2022); see the book Galichon (2018). These works are key inspirations for the present paper, which aims at providing more theoretical understanding in the case of inverse OT (as detailed next).

**Metric Learning.** The estimation of some metrics from pairwise interactions (either positive or negative) falls into the classical field of metric learning in ML, and we refer to the monograph Bellet et al. (2013) for more details. In contrast to the inverse OT (iOT) problem considered in this paper, classical metric learning is more straightforward, as no global matching between sets of points is involved. This allows the metric to be directly optimized, while the iOT problem necessitates some form of bilevel optimization. Similarly to our approach, since the state space is typically continuous, it is necessary to restrict the class of distances to render the problem tractable. The common option, which we also adopt in our paper to exemplify our findings, is to consider the class of Mahalanobis distances. These distances generalize the Euclidean distance and are equivalent to computing a vectorial embedding of the data points. See, for instance, Xing et al. (2002); Weinberger et al. (2006); Davis & Dhillon (2008).

**OT Ground Metric Learning.** The problem of estimating the ground cost driving OT in a supervised manner was first addressed by Cuturi & Avis (2014). Unlike methods that have access to pairs of samples, the ground metric learning problem requires pairs of probability distributions and then evolves into a classical metric learning problem, but within the OT space. The class of ground metrics can be constrained, for example, by utilizing Mahalanobis Wang & Guibas (2012); Xu et al. (2018); Kerdoncuff et al. (2021) or geodesic distances Heitz et al. (2021), to devise more efficient learning schemes. The study Zen et al. (2014) conducts ground metric learning and matrix factorization simultaneously, finding applications in NLP Huang et al. (2016). It is noteworthy that ground metric learning can also be linked to generative models through adversarial training Genevay et al. (2018) and to robust learning Paty & Cuturi (2020) by maximizing the cost to render the OT distance as discriminative as possible.

**Inverse Optimal Transport.** The inverse optimal transport problem (iOT) can be viewed as a specific instance of ground metric learning, where one aims to infer the ground cost from partial observations of the (typically entropically regularized) optimal transport coupling. This problem was first formulated and examined by Dupuy and Galichon Dupuy & Galichon (2014) over a discrete space (also see Galichon & Salanié (2022) for a more detailed analysis), making the fundamental remark that the maximum likelihood estimator amounts to solving a convex problem. The mathematical properties of the iOT problem for discrete space (i.e., direct computation of the cost between all pairs of points) are explored in depth in Chiu et al. (2022), studying uniqueness (up to trivial ambiguities) and stability to pointwise noise. Note that our theoretical study differs fundamentally as we focus on continuous state spaces. This "continuous" setup assumes access only to a set of couples, corresponding to matches (or links) presumed to be drawn from an OT coupling. In this scenario, the iOT is typically an ill-posed problem, and Dupuy et al. (2019); Carlier et al. (2023) propose regularizing the maximum likelihood estimator with either a low-rank (using a nuclear norm penalty) or a sparse prior (using an $\ell^1$ Lasso-type penalty). In our work, we concretely focus on the sparse case, but our theoretical treatment of the iOT could be extended to general structured convex regularization, along the lines of Vaiter et al. (2015). While not the focus of our paper, it is noteworthy that these works also propose efficient large-scale, non-smooth proximal solvers to optimize the penalized maximum likelihood functional, and we refer to Ma et al. (2020) for an efficient solver without inner loop calls to Sinkhorn's algorithm. This approach was further refined in Stuart & Wolfram (2020), deriving it from a Bayesian interpretation, enabling the use of MCMC methods to sample the posterior instead of optimizing a pointwise estimate (as we consider here). They also propose parameterizing cost functions as geodesic distances on graphs (while we consider only linear models to maintain convexity). An important application of iOT to ML, explored by Li et al. (2019), is to perform link prediction by solving new OT problems once the cost has been estimated from the observed couplings. Another category

of ML application of iOT is representation learning (learning embeddings of data into, e.g., an Euclidean space) from pairwise interactions, as demonstrated by Shi et al. (2023), which recasts contrastive learning as a specific instance of iOT.

**Inverse problems and model selection.** The iOT problem is formally a bilevel problem, as the observation model necessitates solving an OT problem as an inner-level program Colson et al. (2005) – we refer to Eisenberger et al. (2022) for a recent numerical treatment of bilevel programming with entropic OT. The iOT can thus be conceptualized as an "inverse optimization" problem Zhang & Liu (1996); Ahuja & Orlin (2001), but with a particularly favorable structure, allowing it to be recast as a convex optimization. This provides the foundation for a rigorous mathematical analysis of performance, as we propose in this paper. The essence of our contributions is a theoretical examination of the recoverability of the OT cost from noisy observations, and particularly, the robustness to noise of the sparse support of the cost (for instance, viewed as a symmetric matrix for Mahalanobis norms). There exists a rich tradition of similar studies in the fields of inverse problem regularization and model selection in statistics. The most prominent examples are the sparsistency theory of the Lasso Tibshirani (1996) (least square regularized by $\ell^1$), which culminated in the theory of compressed sensing Candès et al. (2006). These theoretical results are predicated on a so-called "irrepresentability condition" Zhao & Yu (2006), which ensures the stability of the support. While our analysis is grounded in similar concepts (in particular, we identify the corresponding irrepresentability condition for the iOT), the iOT inverse problem is fundamentally distinct due to the differing observation model (it corresponds to a sampling process rather than the observation of a vector) and the estimation process necessitates solving a linear program of potentially infinite dimension (in the limit of a large number of samples). This mandates a novel proof strategy, which forms the core of our mathematical analysis.

## 1.2 Contributions

This paper proposes the first mathematical analysis of the performance of regularized iOT estimation, focusing on the special case of sparse $\ell^1$ regularization (the $\ell^1$-iOT method). We begin by deriving the customary "irrepresentability condition" of the iOT problem, rigorously proving that it is well-defined. This condition interweaves the properties of the Hessian of the maximum likelihood functional with the sparse support of the sought-after cost. The main contribution of this paper is Theorem 5, which leverages this abstract irrepresentability condition to ensure sparsistency of the $\ell^1$-iOT method. This relates to the robust estimation of the cost and its support in some linear model, assuming the number $n$ of samples is large enough. Specifically, we demonstrate a sample complexity of $n^{-1/2}$. Our subsequent sets of contributions are centered on the case of matching between samples of Gaussian distributions. Herein, we illustrate in Lemma 7 how to compute the irrepresentability condition in closed form. This facilitates the examination of how the parameters of the problem, particularly regularization strength and the covariance of the distributions, influence the success and stability of iOT. We further explore the limiting cases of small and large entropic regularization, revealing in Proposition 8 and Proposition 9 that iOT interpolates between the graphical lasso (to estimate the graph structure of the precision matrix) and a classical lasso. This sheds light on the connection between iOT and graph estimation procedures. Simple synthetic numerical explorations in Section 5.2 further provide intuition about how $\varepsilon$ and the geometry of the graph associated with a sparse cost impact sparsistency. As a minor numerical contribution, we present in Appendix F a large-scale $\ell^1$-iOT solver, implemented in JAX and distributed as open-source software.

## 2 Inverse optimal transport

**The forward problem** Given probability distributions $\alpha \in \mathscr{P}(\mathscr{X})$, $\beta \in \mathscr{P}(\mathscr{Y})$ and cost function $c : \mathscr{X} \times \mathscr{Y} \to \mathbb{R}$, the entropic optimal transport problem seeks to compute a coupling density

$$\text{Sink}(c, \varepsilon) \triangleq \underset{\pi \in \mathscr{U}(\alpha, \beta)}{\arg\max} \langle c, \pi \rangle - \frac{\varepsilon}{2} \text{KL}(\pi | \alpha \otimes \beta) \quad \text{where} \quad \langle c, \pi \rangle \triangleq \int c(x, y) d\pi(x, y), \tag{1}$$

where $\text{KL}(\pi | \xi) \triangleq \int \log(d\pi / d\xi) d\pi - \int d\pi$ and $\mathscr{U}(\alpha, \beta)$ is the space of all probability measures $\pi \in \mathscr{P}(\mathscr{X} \times \mathscr{Y})$ with marginals $\alpha, \beta$.

**The inverse problem**  The inverse optimal transport problem seeks to recover the cost function $c$ given an approximation $\hat{\pi}_n$ of the probability coupling $\hat{\pi} \triangleq \mathrm{Sink}(c, \varepsilon)$. A typical setting is an empirical probability coupling $\hat{\pi}_n = \frac{1}{n} \sum_{i=1}^n \delta_{(x_i, y_i)}$ where $(x_i, y_i)_{i=1}^n \overset{iid}{\sim} \hat{\pi}$. See Section 2.2.

**The loss function**  The iOT problem has been proposed and studied in a series of papers, see Section 1.2. The approach is typically to consider some linear parameterization [1] of the cost $c_A(x, y)$ by some parameter $A \in \mathbb{R}^s$. The key observation of Dupuy et al. (2019) is that the negative log-likelihood of $\hat{\pi}$ at parameter value $A$ is given by

$$\mathscr{L}(A, \hat{\pi}) \triangleq -\langle c_A, \hat{\pi} \rangle + W_{\hat{\pi}}(A) \quad \text{where} \quad W_{\hat{\pi}}(A) \triangleq \sup_{\pi \in \mathscr{U}(\hat{\alpha}, \hat{\beta})} \langle c_A, \pi \rangle - \frac{\varepsilon}{2} \mathrm{KL}(\pi | \hat{\alpha} \otimes \hat{\beta}),$$

and $\hat{\alpha}$, $\hat{\beta}$ are the marginals of $\hat{\pi}$. For ease of notation, unless stated otherwise, we write $W = W_{\hat{\pi}}$. So, computing the maximum likelihood estimator $A$ for the cost corresponds to minimizing the *convex* 'loss function' $A \mapsto \mathscr{L}(A, \hat{\pi})$, which, by regarding $W_{\hat{\pi}}$ as a convex conjugate, can be seen as an instance of a Fenchel-Young loss, a family of losses proposed in Blondel et al. (2020) . We write the parameterization as

$$\Phi : A \in \mathbb{R}^s \mapsto c_A = \sum_{j=1}^s A_j \mathbf{C}_j, \quad \text{where} \quad \mathbf{C}_j \in \mathscr{C}(\mathscr{X} \times \mathscr{Y}).$$

A relevant example are quadratic loss functions, so that for $\mathscr{X} \subset \mathbb{R}^{d_1}$, $\mathscr{Y} \subset \mathbb{R}^{d_2}$, given $A \in \mathbb{R}^{d_1 \times d_2}$, $c_A(x, y) = x^\top A y$. In this case, $s = d_1 d_2$ and for $k = (i, j) \in [d_1] \times [d_2]$, $\mathbf{C}_k(x, y) = x_i y_j$.

**$\ell_1$-iOT**  To handle the presence of noisy data (typically coming from the sampling process), various regularization approaches have been proposed. In this work, we focus on the use of $\ell_1$-regularization Carlier et al. (2023) to recover sparse parametrizations:

$$\underset{A}{\arg\min} \mathscr{F}(A), \quad \text{where} \quad \mathscr{F}(A) \triangleq \lambda \|A\|_1 + \mathscr{L}(A, \hat{\pi}). \tag{iOT$-\ell_1(\hat{\pi})$}$$

**Kantorovich formulation**  Note that $W(A)$ is defined via a concave optimization problem and by Fenchel duality, one can show that (iOT$-\ell_1(\hat{\pi})$) has the following equivalent Kantorovich formulation Carlier et al. (2023):

$$\underset{A, f, g}{\arg\min} \mathscr{K}(A, f, g), \quad \text{where} \quad \mathscr{K}(A, f, g) \triangleq \mathscr{J}(A, f, g) + \lambda \|A\|_1, \quad \text{and} \tag{$\mathscr{K}_\infty$}$$

$$\mathscr{J}(A, f, g) \triangleq -\int \left( f(x) + g(y) + \Phi A(x, y) \right) d\hat{\pi}(x, y) + \frac{\varepsilon}{2} \int \exp \left( \frac{2(f(x) + g(y) + \Phi A(x, y))}{\varepsilon} \right) d\alpha(x) d\beta(y).$$

Based on this formulation, various algorithms have been proposed, including alternating minimization with proximal updates Carlier et al. (2023). Section F details a new large scale solver that we use for the numerical simulations.

## 2.1  INVARIANCES AND ASSUMPTIONS

*Assumption* 1.  We first assume that $\mathscr{X}$ and $\mathscr{Y}$ are compact.

Note that $\mathscr{J}$ has the translation invariance property that for any constant function $u$, $\mathscr{J}(f + u, g - u, A) = \mathscr{J}(f, g, A)$, so to remove this invariance, throughout, we restrict the optimization of $(\mathscr{K}_\infty)$ to the set

$$\mathscr{S} \triangleq \left\{ (A, f, g) \in \mathbb{R}^s \times L^2(\alpha) \times L^2(\beta) \, ; \, \int g(y) d\beta(y) = 0 \right\}. \tag{2}$$

Next, we make some assumptions on the cost to remove invariances in the iOT problem.

*Assumption* 2 (Assumption on the cost).  (i)  $\mathbb{E}_{(x, y) \sim \alpha \otimes \beta}[\mathbf{C}(x, y) \mathbf{C}(x, y)^\top] \succeq \mathrm{Id}$ is invertible.

(ii)  $\|\mathbf{C}(x, y)\| \leq 1$ for $\alpha$ almost every $x$ and $\beta$-almost every $y$.

---

[1] In this work, we restrict to linear parameterizations, although the same loss function can also be applied to learn costs with nonlinear parameterization, e.g. via neural networks Ma et al. (2020).

(iii) for all $k$, $\int \mathbf{C}_k(x, y) d\alpha(x) = 0$ for $\beta$-a.e. $y$ and $\int \mathbf{C}_k(x, y) d\beta(y) = 0$ for $\alpha$-a.e. $x$.

Under these assumptions, it can be shown that iOT has a unique solution (see remark after Proposition 2). Assumption 2 (i) is to ensure that $c_A = \Phi A$ is uniquely determined by $A$. Assumption 2 (ii) is without loss of generality, since we assume that $\alpha, \beta$ are compactly supported, so this holds up to a rescaling of the space. Assumption 2 (iii) is to handle the invariances pointed out in Carlier et al. (2023) and Ma et al. (2020):

$$\mathcal{J}(A, f, g) = \mathcal{J}(A', f', g') \iff c_A + (f \oplus g) = c_{A'} + (f' \oplus g'). \tag{3}$$

As observed in Carlier et al. (2023), any cost can be adjusted to fit this assumption: one can define

$$\tilde{\mathbf{C}}_k(x, y) = \mathbf{C}_k(x, y) - u_k(x) - v_k(y)$$

where $u_k(x) = \int \mathbf{C}_k(x, y) d\beta(y)$ and $v_k(y) = \int \mathbf{C}_k(x, y) d\alpha(x) - \int \mathbf{C}_k(x, y) d\alpha(x) d\beta(y)$. Letting $\tilde{\Phi} A = \sum_k A_k \tilde{\mathbf{C}}_k$, we have $\left( \left( f - \sum_k A_k u_k \right) \oplus \left( g - \sum_k A_k v_k \right) \right) + \tilde{\Phi} A = (f \oplus g) + \Phi A$. So optimization with the parametrization $\Phi$ is equivalent to optimization with $\tilde{\Phi}$.

NB: For the quadratic cost $c_k(x, y) = x_i y_j$ for $k = (i, j)$, condition (iii) corresponds to *recentering* the data points, and taking $x \mapsto x - \int x d\alpha(x)$ and $y \mapsto y - \int y d\beta(y)$. Condition (ii) holds if $\|x\| \vee \|y\| \leqslant 1$ for $\alpha$-a.e. $x$ and $\beta$-a.e. $y$. Condition (i) corresponds to invertibility of $\mathbb{E}_\alpha[x x^\top] \otimes \mathbb{E}_\beta[y y^\top]$.

## 2.2 THE FINITE SAMPLE PROBLEM

In practice, we do not observe $\hat{\pi}$ but $n$ data-points $(x_i, y_i) \overset{iid}{\sim} \hat{\pi}$ for $i = 1, \ldots, n$, where $\hat{\pi} = \text{Sink}(c_{\hat{A}}, \varepsilon)$. To recover $\hat{A}$, we plug into iOT$-\ell_1$ the empirical measure $\hat{\pi}_n = \frac{1}{n} \sum_{i=1}^n \delta_{x_i, y_i}$ and consider the estimator

$$A_n \in \underset{A}{\arg\min} \, \lambda \|A\|_1 + \mathcal{L}(A, \hat{\pi}_n), \tag{iOT$-\ell_1(\hat{\pi}_n)$}$$

As in section 2.1, to account for the invariance (3), when solving (iOT$-\ell_1(\hat{\pi}_n)$), we again centre the cost parameterization such that for all $i$, $\sum_{i=1}^n \mathbf{C}_k(x_i, y_j) = 0$ and for all $j$, $\sum_{j=1}^n \mathbf{C}_k(x_i, y_j) = 0$. Note also that iOT$-\ell_1(\hat{\pi}_n)$ can be formulated entirely in finite dimensions. See Appendix B for details.

## 3 THE CERTIFICATE FOR SPARSISTENCY

In this section, we consider the problem (iOT$-\ell_1(\hat{\pi})$) with full data $\hat{\pi}$ and present a sufficient condition for support recovery, that we term *non-degeneracy of the certificate*. For simplicity of notation, throughout this section denote $W \triangleq W_{\hat{\pi}}$. Under non-degeneracy of the certificate we obtain support recovery as stated in Theorem 3, a known result whose proof can be found *e.g.* in Lee et al. (2015). This condition can be seen as a generalization of the celebrated *Lasso's Irrepresentability condition* (see *e.g.* Hastie et al. (2015)) – Lasso corresponds to having a quadratic loss instead of $\mathcal{L}(A, \hat{\pi})$, thus $\nabla_A^2 W(A)$ in the definition below reduces to a matrix. In what follows, we denote $u_I \triangleq (u_i)_{i \in I}$ and $U_{I,J} \triangleq (U_{i,j})_{i \in I, j \in J}$ the restriction operators.

**Definition 1.** *The* certificate *with respect to $A$ and support $I = \{i : A_i \neq 0\}$ is*

$$z_A^* \triangleq \nabla^2 W(A)_{(:, I)} (\nabla^2 W(A)_{(I, I)})^{-1} \text{sign}(A)_I. \tag{C}$$

*We say that it is non-degenerate if $\|(z_A^*)_{I^c}\|_\infty < 1$.*

The next proposition, whose proof can be found in Appendix C.1, shows that the function $W(A)$ is twice differentiable, thus ensuring that C is well defined.

**Proposition 2.** *$A \mapsto W(A)$ is twice differentiable, strictly convex, with gradient and Hessian*

$$\nabla_A W(A) = \Phi^* \pi_A, \quad \nabla_A^2 W(A) = \Phi^* \frac{\partial \pi_A}{\partial A}(x, y)$$

*where $\pi_A$ is the unique solution to (1) with cost $c_A = \Phi A$.*

We remark that strict convexity of $W$ implies that any solution of (iOT$-\ell_1(\hat{\pi})$) must be unique, and by $\Gamma$-convergence, solutions $A^\lambda$ to (iOT$-\ell_1(\hat{\pi})$) converge to $\hat{A}$ as $\lambda \to 0$. The next theorem, a well-known result (see (Lee et al., 2015, Theorem 3.4) for a proof), shows that support recovery can be characterized via $z_{\hat{A}}^*$

**Theorem 3.** *Let $\hat{\pi} = \text{Sink}(c_{\widehat{A}}, \varepsilon)$. If $z_{\widehat{A}}^*$ is non-degenerate, then for all $\lambda$ sufficiently small, the solution $A^\lambda$ to ($\text{iOT}-\ell_1(\hat{\pi})$) is sparsistent with $\|A^\lambda - \widehat{A}\| = \mathcal{O}(\lambda)$ and $\text{Supp}(A^\lambda) = \text{Supp}(\widehat{A})$.*

### 3.1 Intuition behind the certificate (C)

**Implication of the non-degeneracy condition** The non-degeneracy condition is widely studied for the Lasso problem Hastie et al. (2015). Just as for the Lasso, the closer $\left|(z_{\widehat{A}}^*)_i\right|$ is to 1, the more unstable is this coefficient, and if $\left|(z_{\widehat{A}}^*)_i\right| > 1$ then one cannot expect to recover this coefficients when there is noise. Note also that in the Lasso, the pre-certificate formula in (C) roughly the *correlation* between coefficient inside and outside the support $I$. In our case, the loss is more complex and this *correlation* is measured according to the hessian of $W$, which integrates the curvature of the loss. This curvature formula is however involved, so to gain intuition, we perform a detailed analysis in the $\varepsilon \to 0$ and $\varepsilon - > \infty$ for the Gaussian setting where it becomes much simpler (see Section 5.1).

**Link to optimality conditions** The certificate $z_{\widehat{A}}^*$ can be seen as the *limit optimality condition* for the optimization problem ($\text{iOT}-\ell_1(\hat{\pi})$) as $\lambda \to 0$: by the first order optimality condition to ($\text{iOT}-\ell_1(\hat{\pi})$), $A^\lambda$ is a solution if and only if $z^\lambda \triangleq -\frac{1}{\lambda}\nabla L(A^\lambda) \in \partial\|A_\lambda\|_1$, where the subdifferential for the $\ell_1$ norm has the explicit form $\partial\|A\|_1 = \{z \; ; \; \|z\|_\infty \leqslant 1, \; \forall i \in \text{Supp}(A), z_i = \text{sign}(A_i)\}$. It follows $z^\lambda$ can be seen as a *certificate* for the support of $A^\lambda$ since $\text{Supp}(A^\lambda) \subseteq \{i \; ; \; |z_i^\lambda| = 1\}$. To study the support behavior of $A^\lambda$ for small $\lambda$, it is therefore interesting to consider the limit of $z^\lambda$ as $\lambda \to 0$. Its limit is precisely the subdifferential element with the minimal norm and coincides with (C) under the nondegeneracy condition:

**Proposition 4.** *Let $z^\lambda \triangleq -\frac{1}{\lambda}\nabla L(A^\lambda)$ where $A^\lambda$ solves ($\text{iOT}-\ell_1(\hat{\pi})$). Then,*

$$\lim_{\lambda \to 0} z^\lambda = z_{\widehat{A}}^{\min} \triangleq \arg\min_z \left\{ \langle z, (\nabla^2 W(\widehat{A}))^{-1} z \rangle_F \; ; \; z \in \partial\|\widehat{A}\|_1 \right\} \tag{MNC}$$

*Moreover, if $z_{\widehat{A}}^*$ is non-degenerate, then $z_{\widehat{A}}^{\min} = z_{\widehat{A}}^*$.*

## 4 Sample complexity bounds

Our main contribution shows that (C) is a certificate for sparsistency under sampling noise:

**Theorem 5.** *Let $\hat{\pi} = \text{Sink}(c_{\widehat{A}}, \varepsilon)$. Suppose that the certificate $z_{\widehat{A}}^*$ is non-degenerate. Let $\delta > 0$. Then, for all sufficiently small regularization parameters $\lambda$ and sufficiently many number of samples $n$,*

$$\lambda \lesssim 1 \quad and \quad \max\left(\exp(C\|\widehat{A}\|_1/\varepsilon)\sqrt{\log(1/\delta)}\lambda^{-1}, \sqrt{\log(2s)}\right) \lesssim \sqrt{n},$$

*for some constant $C > 0$, with probability at least $1 - \delta$, the minimizer $A_n$ to $(\mathscr{P}_n)$ is sparsistent with $\widehat{A}$ with $\text{Supp}(A_n) = \text{Supp}(\widehat{A})$ and $\|A_n - \widehat{A}\|_2 \lesssim \lambda + \sqrt{\exp(C\|\widehat{A}\|_1/\varepsilon)\log(1/\delta)n^{-1}}$.*

**Main idea behind Theorem 5** We know from Theorem 3 that there is some $\lambda_0 > 0$ such that for all $\lambda \leqslant \lambda_0$, the solution to $(\mathscr{K}_\infty)$ has the same support as $\widehat{A}$. To show that the finite sample problem also recovers the support of $\widehat{A}$ when $n$ is sufficiently large, we **fix** $\lambda \in (0, \lambda_0]$ and consider the setting where the observations are iid samples from the coupling measure $\hat{\pi}$. We will derive convergence bounds for the primal and dual solutions as the number of samples $n$ increases. Let $(A_\infty, f_\infty, g_\infty)$ minimise $(\mathscr{K}_\infty)$. Denote $F_\infty = (f_\infty(x_i))_{i \in [n]}$, $G_\infty = (g_\infty(y_j))_{j \in [n]}$ and

$$P_\infty = \frac{1}{n^2}(p_\infty(x_i, y_j))_{i,j \in [n]}, \quad \text{where} \quad p_\infty(x, y) = \exp\left(\frac{2}{\varepsilon}\left(\Phi A_\infty(x, y) + f_\infty(x) + g_\infty(y)\right)\right).$$

Let $A_n$ minimize $\text{iOT}-\ell_1(\hat{\pi}_n)$. Then, there exists a probability matrix $P_n \in \mathbb{R}_+^{n \times n}$ and vectors $F_n, G_n \in \mathbb{R}^n$ such that $P_n = \frac{1}{n^2}\exp\left(\frac{2}{\varepsilon}(\Phi_n A_n + F_n \oplus G_n)\right)$. In fact, $(A_n, F_n, G_n)$ minimize the finite dimensional dual problem $(\mathscr{K}_n)$ given in the appendix. Now consider the *certificates*

$$z_\infty = \frac{1}{\lambda}\Phi^*\left(p_\infty \alpha \otimes \beta - \hat{\pi}\right) \quad \text{and} \quad z_n = \frac{1}{\lambda}\Phi_n^*\left(P_n - \hat{P}_n\right).$$

Note that $z_\infty$ and $z_n$ both depend on $\lambda$; the superscript was dropped since $\lambda$ is fixed. Moreover, $z_\infty$ is precisely $z^\lambda$ from Proposition 4. By exploiting strong convexity properties of $\mathscr{J}_n$, one can show the following sample complexity bound on the convergence of $z_n$ to $z_\infty$ (the proof can be found in the appendix):

**Proposition 6.** *Let $n \gtrsim \max\left(\log(1/\delta)\lambda^{-2}, \log(2s)\right)$ for some $\delta > 0$. For some constant $C > 0$, with probability at least $1 - \delta$, $\|z_\infty - z_n\|_\infty \lesssim \exp(C\|\widehat{A}\|_1 / \varepsilon)\log(1/\delta)\lambda^{-1} n^{-\frac{1}{2}}$ and*

$$\|A_n - A_\infty\|_2^2 + \frac{1}{n}\sum_i (F_n - F_\infty)_i^2 + \frac{1}{n}\sum_j (G_n - G_\infty)_j^2 \lesssim \varepsilon^2 \exp(C\|\widehat{A}\|_1 / \varepsilon)\log(1/\delta)n^{-1}.$$

From Proposition 4, for all $\lambda \leq \lambda_0$ for some $\lambda_0$ sufficiently small, the magnitude of $z_\infty$ outside the support of $\widehat{A}$ is less than one. Moreover, the convergence result in Proposition 6 above implies that, for $n$ sufficiently large, the magnitude of $z_n$ outside the support of $\widehat{A}$ is also less than one. Hence, since the set $\{i \; ; \; (z_n)_i = \pm 1\}$ determines the support of $A_n$, we have sparsistency.

## 5 GAUSSIAN DISTRIBUTIONS

To get further insight about the sparsistency property ot iOT, we consider the special case where the source and target distributions are Gaussians, and the cost parametrization $c_A(x, y) = x^\top A y$. To this end, we first derive closed form expressions for the Hessian $\partial_A^2 \mathscr{L}(A) = \nabla_A^2 W(A)$. Given $\alpha = \mathscr{N}(m_\alpha, \Sigma_\alpha)$ and $\beta = \mathscr{N}(m_\beta, \Sigma_\beta)$, it is known (see Bojilov & Galichon (2016)) that the coupling density is also a Gaussian of the form $\pi = \mathscr{N}\left(\binom{m_\alpha}{m_\beta}, \begin{pmatrix} \Sigma_\alpha & \Sigma \\ \Sigma^\top & \Sigma_\beta \end{pmatrix}\right)$ for some $\Sigma \in \mathbb{R}^{d_1 \times d_2}$. In this case, $W$ can be written as an optimization problem over the cross-covariance $\Sigma$ Bojilov & Galichon (2016).

$$W(A) = \sup_{\Sigma \in \mathbb{R}^{d_1 \times d_2}} \langle A, \Sigma \rangle + \frac{\varepsilon}{2}\log\det\left(\Sigma_\beta - \Sigma^\top \Sigma_\alpha^{-1} \Sigma\right), \tag{4}$$

with the optimal solution being precisely the cross-covariance of the optimal coupling $\pi$. In Bojilov & Galichon (2016), the authors provide an explicit formula for the minimizer $\Sigma$ to (4), and, consequently, for $\nabla W(A)$:

$$\Sigma = \Sigma_\alpha A \Delta \left(\Delta A^\top \Sigma_\alpha A \Delta\right)^{-\frac{1}{2}} \Delta - \frac{1}{2}\varepsilon A^{\dagger,\top} \quad \text{where} \quad \Delta \triangleq \left(\Sigma_\beta + \frac{\varepsilon^2}{4} A^\dagger \Sigma_\alpha^{-1} A^{\dagger,\top}\right)^{\frac{1}{2}}. \tag{5}$$

By differentiating the first order condition for $W$, that is $A^\dagger = \varepsilon^{-1}(\Sigma_\beta - \Sigma^\top \Sigma_\alpha^{-1} \Sigma)\Sigma^\dagger \Sigma_\alpha$, Galichon derives the expression for the Hessian in terms of $\Sigma$ in (5):

$$\nabla^2 W(A) = \partial_A \Sigma = \varepsilon \left(\Sigma_\alpha \Sigma^{-1,\top} \otimes \Sigma_\beta \Sigma^{-1} + \mathbb{T}\right)^{-1} \left(A^{-1,\top} \otimes A^{-1}\right), \tag{6}$$

where $\mathbb{T}$ is such that $\mathbb{T}\mathrm{vec}(A) = \mathrm{vec}(A^\top)$. This formula does not hold in when $A$ is rectangular or rank deficient, since $A^\dagger$ is not differentiable. In the following, we derive, via the implicit function theorem, a general formula for $\partial_A \Sigma$ that agrees with that of Galichon in the square invertible case.

**Lemma 7.** *Denoting $\Sigma$ as in (5), one has*

$$\nabla^2 W(A) = \varepsilon \left(\varepsilon^2 (\Sigma_\beta - \Sigma^\top \Sigma_\alpha \Sigma)^{-1} \otimes (\Sigma_\alpha - \Sigma \Sigma_\beta^{-1} \Sigma^\top)^{-1} + (A^\top \otimes A)\mathbb{T}\right)^{-1} \tag{7}$$

This formula given in Lemma 7 provides an explicit expression for the certificate (C).

### 5.1 LIMIT CASES FOR LARGE AND SMALL $\varepsilon$

This section explores the behaviour of the certificate in the large/small $\varepsilon$ limits: Proposition 8 reveals that the large $\varepsilon$ limit coincides with the classical Lasso while Proposition 9 reveals that the small epsilon limit (for symmetric $A > 0$ and $\Sigma_\alpha = \Sigma_\beta = \mathrm{Id}$) coincides with the Graphical Lasso. In the following results, we denote the functional in (iOT$-\ell_1(\hat{\pi})$) with parameters $\lambda$ and $\varepsilon$ by $\mathscr{F}_{\varepsilon,\lambda}(A)$.

**Proposition 8** ($\varepsilon \to \infty$). *Let $\widehat{A}$ be invertible and let $\hat{\pi} = \mathrm{Sink}(c_{\widehat{A}}, \varepsilon)$ be the observed coupling between $\alpha = \mathscr{N}(m_\alpha, \Sigma_\alpha)$ and $\beta = \mathscr{N}(m_\beta, \Sigma_\beta)$. Then,*

$$\lim_{\varepsilon \to \infty} z_\varepsilon = (\Sigma_\beta \otimes \Sigma_\alpha)_{(:,I)}\left((\Sigma_\beta \otimes \Sigma_\alpha)_{(I,I)}\right)^{-1}\mathrm{sign}(\widehat{A})_I. \tag{8}$$

*Moreover, for $\lambda_0 > 0$, given any sequence $(\varepsilon_j)_j$ and $A_j \in \arg\min_A \mathscr{F}_{\varepsilon,\lambda_0/\varepsilon_j}(A)$ with $\lim_{j\to\infty} \varepsilon_j = \infty$, any cluster point of $(A_j)_j$ is in*

$$\underset{A\in\mathbb{R}^{d\times d}}{\arg\min} \lambda_0 \|A\|_1 + \frac{1}{2} \|(\Sigma_\beta^{1/2} \otimes \Sigma_\alpha^{1/2})(A - \widehat{A})\|_F^2 \tag{9}$$

**Interpretation** As $\varepsilon \to \infty$, the KL term forces $\hat\pi$ to be close to the *independent coupling $\alpha \otimes \beta$* and in the limit, iOT is simply a Lasso problem and the limit in (8) is precisely the Lasso certificate Hastie et al. (2015). Here, the cross covariance of $\hat\pi$ satisfies $\varepsilon\Sigma = A + \mathcal{O}(\varepsilon)$ ((53) in the appendix), so for large $\varepsilon$, sparsity in $A$ indicates the independence in the coupling between $\alpha$ and $\beta$.

**Proposition 9** ($\varepsilon \to 0$). *Let $\widehat{A}$ be symmetric positive-definite and let $\hat\pi = \text{Sink}(c_{\widehat{A}}, \varepsilon)$ be the observed coupling between $\alpha = \mathcal{N}(m_\alpha, \text{Id})$ and $\beta = \mathcal{N}(m_\beta, \text{Id})$. Then,*

$$\lim_{\varepsilon\to 0} z_\varepsilon = (\widehat{A}^{-1} \otimes \widehat{A}^{-1})_{(:,I)} \big((\widehat{A}^{-1} \otimes \widehat{A}^{-1})_{(I,I)}\big)^{-1} \text{sign}(\widehat{A})_I. \tag{10}$$

*Let $\lambda_0 > 0$. Then, optimizing over symmetric positive semi-definite matrices, given any sequence $(\varepsilon_j)_j$ and $A_j \in \arg\min_{A\geq 0} \mathscr{F}_{\varepsilon,\lambda_0\varepsilon_j}(A)$ with $\lim_{j\to\infty} \varepsilon_j = 0$, any cluster point of $(A_j)_j$ is in*

$$\underset{A\geq 0}{\arg\min} \lambda_0 \|A\|_1 - \frac{1}{2}\log\det(A) + \frac{1}{2}\langle A, \widehat{A}^{-1}\rangle. \tag{11}$$

**Interpretation** In contrast, $\varepsilon \to 0$, the KL term disappears and the coupling $\hat\pi$ becomes dependent. Naturally, the limit problem is the graphical lasso, typically used to infer conditional independence in graphs (but where covariates can be highly dependent). Note also that the limit (10) is precisely the graphical Lasso certificate Hastie et al. (2015). Here, (Remark 25 in the appendix) one show that for $(x, y) \sim \pi$, the conditional covariance of $x$ conditional on $y$ (and also vice versa) is $\varepsilon A^{-1} + \mathcal{O}(\varepsilon^2)$. Sparsity in $A$ can therefore be viewed as information on conditional independence.

## 5.2 NUMERICAL ILLUSTRATIONS

In order to gain some insight into the impact of $\varepsilon$ and the covariance structure on the efficiency of iOT, we present numerical computations of certificates here. We fix the covariances of the input measures as $\Sigma_\alpha = \Sigma_\beta = \text{Id}_n$, similar results are obtained with different covariance as long as they are not rank-deficient. We consider that the support of the sought-after cost matrix $A = \delta\text{Id}_n + \text{diag}(G1_n) - G \in \mathbb{R}^{n\times n}$ is defined as a shifted Laplacian matrix of some graph adjacency matrix $G$, for a graph of size $n = 80$ (similar conclusions hold for larger graphs). We set the shift $\delta$ to be 10% of the largest eigenvalue of the Laplacian, ensuring that $C$ is symmetric and definite. This setup corresponds to graphs defining positive interactions at vertices and negative interactions along edges. For small $\varepsilon$, adopting the graphical lasso interpretation (as exposed in Section 5.1) and interpreting $C$ as a precision matrix, this setup corresponds (for instance, for a planar graph) to imposing a spatially smoothly varying covariance $C^{-1}$. Figure 1 illustrates how the value of the certificates $z_{i,j}$ evolves depending on the indexes $(i, j)$ for three types of graphs (circular, planar, and Erdős–Rényi with a probability of edges equal to 0.1), for several values of $\varepsilon$. By construction, $z_{i,i} = 1$ and $z_{i,j} = -1$ for $(i, j)$ connected by the graph. For $z$ to be non-degenerate, it is required that $|z_{i,j}| < 1$, as $i$ moves away from $j$ on the graph. For the circular and planar graphs, the horizontal axis represents the geodesic distance $d_{\text{geod}}(i, j)$, demonstrating how the certificates become well-behaved as the distance increases. The planar graph displays envelope curves showing the range of values of $z$ for a fixed value of $d_{\text{geod}}(i, j)$, while this is a single-valued curve for the circular graph due to periodicity. For the Erdős–Rényi graph, to better account for the randomness of the certificates along the graph edges, we display the histogram of the distribution of $|z_{i,j}|$ for $d_{\text{geod}}(i, j) = 2$ (which represents the most critical set of edges, as they are the most likely to be large). All these examples show the same behavior, namely, that increasing $\varepsilon$ improves the behavior of the certificates (which is in line with the stability analysis of Section 5.1, since (8) implies that for large $\varepsilon$, the certificate is trivially non-degenerate whenever $\Sigma_\alpha, \Sigma_\beta$ are diagonal), and that pairs of vertices $(i, j)$ connected by a small distance $d_{\text{geod}}(i, j)$ are the most likely to be degenerate. This suggests that they will be inaccurately estimated by iOT for small $\varepsilon$.

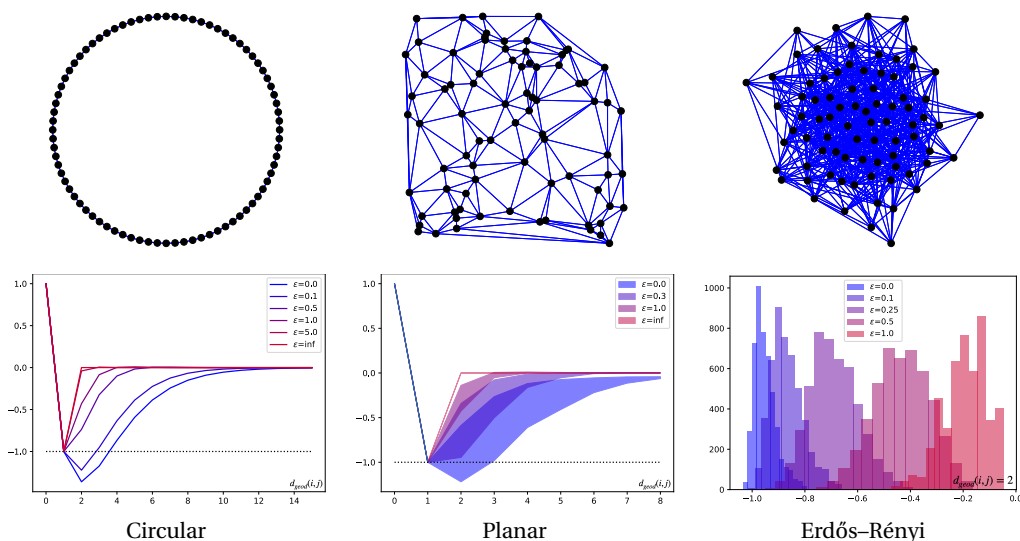

Figure 1: Display of the certificate values $z_{i,j}$ for three types of graphs, for varying $\varepsilon$. Left, middle: plotted as a function of the geodesic distance $d_{\text{geod}}(i, j)$ on the $x$-axis. Right: histogram of $z_{i,j}$ for $(i, j)$ at distance $d_{\text{geod}}(i, j) = 2$.

Figure 2 displays the recovery performances of $\ell^1$–iOT for the circular graph shown on the left of Figure 1 (similar results are obtained for the other types of graph topologies). These numerical simulations are obtained using the large-scale iOT solver, which we detail in Appendix F. The performance is represented using the number of inaccurately estimated coordinates in the estimated cost matrix $C$, so a score of 0 means a perfect estimation of the support (sparsistency is achieved). For $\varepsilon = 0.1$, sparsistency cannot be achieved, aligning with the fact that the certificate $z$ is degenerate as depicted in the previous figure. In sharp contrast, for larger $\varepsilon$, sparsistency can be attained as soon as the number of samples $N$ is sufficiently large, which also aligns with our theory of sparsistency of $\ell^1$–iOT since the certificate is guaranteed to be non-degenerate in the large $\varepsilon$ setting.

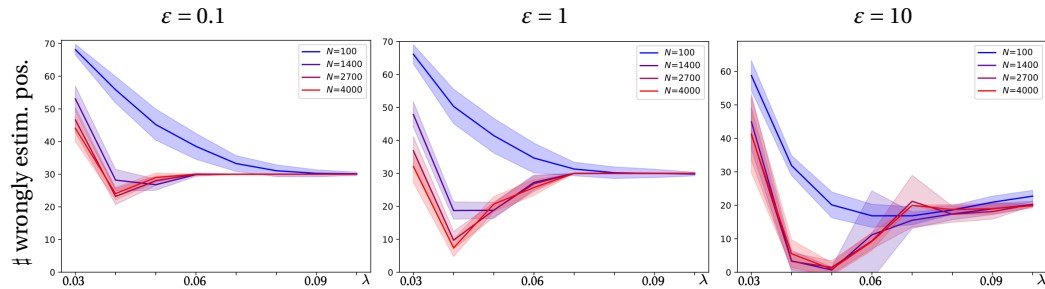

Figure 2: Recovery performance (number of wrongly estimated position) of $\ell^1$–iOT as a function of $\lambda$ for three different values of $\varepsilon$.

Figure 3 in the appendix displays the certificate values in the case of a non-symmetric planar graph. The graph is obtained by deleting all edges on a planar graph with $i < j$. We plot the certificate values as a function of geodesic distances $d_{\text{geod}}(i, j)$ of the symmetrized graph. The middle plot shows the certificate values on $i \geqslant j$ (where the actual edges are constrained to be and nondegeneracy requires values smaller than 1 in absolute value for $d_{\text{geod}}(i, j) \geqslant 2$). The right plot shows the certificate values on $i \leqslant j$ where there are no edges and for nondegeneracy, one expects values smaller than 1 in absolute value for $d_{\text{geod}}(i, j) \geqslant 1$. Observe that here, the certificate is degenerate for small values of $d_{\text{geod}}(i, j)$ when $\varepsilon = 0$, meaning that the problem is unstable at the "ghost" symmetric edges. As $\varepsilon \to \infty$, the certificate becomes non-degenerate.

CONCLUSION

In this paper, we have proposed the first theoretical analysis of the recovery performance of $\ell^1$-iOT. Much of this analysis can be extended to more general convex regularizers, such as the nuclear norm to promote low-rank Euclidean costs, for instance. Our analysis and numerical exploration support the conclusion that iOT becomes ill-posed and fails to maintain sparsity for overly small $\varepsilon$. When approached from the perspective of graph estimations, unstable indices occur at smaller geodesic distances, highlighting the geometric regularity of iOT along the graph geometry.

ACKNOWLEDGEMENTS

The work of G. Peyré was supported by the European Research Council (ERC project NORIA) and the French government under management of Agence Nationale de la Recherche as part of the "Investissements d'avenir" program, reference ANR19-P3IA-0001 (PRAIRIE 3IA Institute).

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

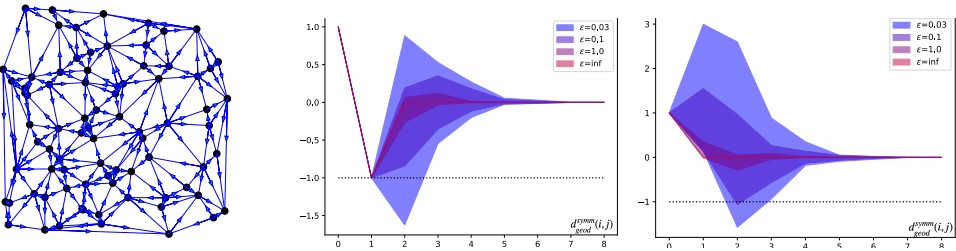

Figure 3: Display of certificate values for a non-symmetric planar graph, for varying $\varepsilon$ with edges only for $i > j$. Middle/Right: plots of the certificate values as a function of the *geodesic distance* $d_{geod}(i, j)$ *of the symmetrized graph*. The middle plot show the values when restricted to $i \geqslant j$. The right plot shows the values restricted to $i \leqslant j$ (where there are no edges).

## A  INTERPRETATIONS OF THE LOSS FUNCTION

As mentioned in Section 2, the loss $\mathscr{L}(A, \hat{\pi})$ can be recovered via maximum likelihood (ML) estimation (Dupuy et al., 2019, Proposition 1) and be regarded as an instance of a family of losses called Fenchel-Young losses. For the sake of completeness, this section explains this in more detail.

### A.1  MAXIMUM LIKELIHOOD INTERPRETATION

The map $A \to \mathrm{Sink}(c_A, \varepsilon)$, where $\mathrm{Sink}(\cdot, \cdot)$ is defined in Section 2, can be seen as parameterizing a set of measures which are absolutely continuous with respect to $\alpha \otimes \beta$. By the standard duality result (see Nutz (2021)):

1) The density of $\mathrm{Sink}(c_A, \varepsilon)$ is given by $d\mathrm{Sink}(c_A, \varepsilon)/d(\alpha \otimes \beta) = \exp(c_A + f_A + g_A)$, where $f_A$ and $g_A$ solve the dual problem, *i.e.*, $\sup_{f,g} \int f \, d\alpha + \int g \, d\beta - \int \exp(c_A + f + g) \, d(\alpha \otimes \beta) + 1$;

2) The values of the primal and dual problems agree and are equal to $-\int f_A \, d\alpha - \int g_A \, d\beta$.

Combining these two facts we obtain that

$$-\log\left(\frac{d\mathrm{Sink}(c_A, \varepsilon)}{d(\alpha \otimes \beta)}\right) = -c_A - f_A - g_A = -c_A + \sup_{\pi \in \mathscr{U}(\alpha,\beta)} \langle c_A, \pi \rangle - \frac{\varepsilon}{2} \mathrm{KL}(\pi | \alpha \otimes \beta) = -c_A + W_{\hat{\pi}}(A),$$

where $W_{\hat{\pi}}$ is defined in as in Section 2. Finally, taking expectation with respect to $\hat{\pi}$ yields

$$\mathbb{E}_{(x,y)\sim\hat{\pi}}\left[-\log\left(\frac{d\mathrm{Sink}(c_A, \varepsilon)}{d(\alpha \otimes \beta)}\right)\right] = \mathbb{E}_{(x,y)\sim\hat{\pi}}\left[-c_A\right] + \mathbb{E}_{(x,y)\sim\hat{\pi}}\left[W_{\hat{\pi}}(A)\right] = -\langle c_A, \hat{\pi} \rangle + W_{\hat{\pi}}(A) = \mathscr{L}(A, \hat{\pi}),$$

thus establishing the connection with ML estimation.

**Bilevel interpretation**   Note also that

$$\underset{A}{\arg\min}\, \mathscr{L}(A, \hat{\pi}) = \underset{A}{\arg\min} -\langle c_A, \hat{\pi} - \pi_A \rangle - \frac{\varepsilon}{2} \mathrm{KL}(\pi_A | \hat{\alpha} \otimes \hat{\beta})$$

$$= \underset{A}{\arg\min} -\langle c_A, \hat{\pi} - \pi_A \rangle - \frac{\varepsilon}{2} \mathrm{KL}(\pi_A | \hat{\alpha} \otimes \hat{\beta}) + \frac{\varepsilon}{2} \mathrm{KL}(\hat{\pi} | \hat{\alpha} \otimes \hat{\beta})$$

where $\pi_A = \arg\min_{\pi \in \mathscr{U}(\hat{\alpha},\hat{\beta})} \langle c_A, \pi \rangle - \mathrm{KL}(\pi | \hat{\alpha} \otimes \hat{\beta})$. Since $\hat{\pi}, \pi_A$ have the same marginals,

$$\langle c_A, \hat{\pi} - \pi_A \rangle = \langle c_A + f_A + g_A, \hat{\pi} - \pi_A \rangle = \frac{\varepsilon}{2} \langle \log(\pi_A/(\hat{\alpha} \otimes \hat{\beta})), \hat{\pi} - \hat{\pi}_A \rangle.$$

We therefore have

$$\arg\min_A \mathscr{L}(A, \hat{\pi}) = \arg\min_A \mathrm{KL}(\hat{\pi} | \pi_A), \quad \text{where} \quad \pi_A = \arg\min_{\pi \in \mathscr{U}(\hat{\alpha},\hat{\beta})} \langle c_A, \pi \rangle - \mathrm{KL}(\pi | \hat{\alpha} \otimes \hat{\beta}).$$

## A.2 FENCHEL-YOUNG LOSS INTERPRETATION

It is argued in Blondel et al. (2020) that, associated to a prediction rule of the form

$$\hat{y}(\theta) = \arg\max_{\mu \in \mathrm{dom}(\Omega)} \langle \theta, \mu \rangle - \Omega(\mu),$$

there is a natural loss function that the authors term Fenchel-Young loss and that is given by

$$L_\Omega(\theta; y) \triangleq \Omega^*(\theta) + \Omega(y) - \langle \theta, y \rangle.$$

With this in mind, let $\Omega(\pi) = \mathrm{KL}(\pi | \alpha \otimes \beta) + \iota_{\mathscr{U}(\alpha,\beta)}(\pi)$ where $\iota_{\mathscr{U}(\alpha,\beta)}(\cdot)$ is the indicator function of $\mathscr{U}(\alpha,\beta)$, and note that

$$\mathscr{L}(A, \hat{\pi}) = L_\Omega(c_A; \hat{\pi}) - \Omega(\hat{\pi}).$$

Since, for the purposes of minimization with respect to $A$, the term $\Omega(\hat{\pi})$ is irrelevant, we see that finding a minimizer of (iOT$-\ell_1(\hat{\pi})$) amounts to finding a minimizer of the $l_1$ regularized Fenchel-Young loss associated with $\mathrm{KL}(\cdot | \alpha \otimes \beta) + \iota_{\mathscr{U}(\alpha,\beta)}(\cdot)$.

## B THE FINITE SAMPLE PROBLEM

As mention in Section 2.2, we do not observe the full coupling $\hat{\pi} = \mathrm{Sink}(c_{\hat{A}}, \varepsilon)$, but only the the empirical measure $\hat{\pi}_n = \frac{1}{n} \sum_{i=1}^n \delta_{x_i, y_i}$ with $(x_i, y_i) \overset{iid}{\sim} \hat{\pi}$. We show here that the problem iOT$-\ell_1(\hat{\pi}_n)$ can be formulated entirely in finite dimensions as follows. Let

$$H(P|Q) \triangleq \sum_{i,j} P_{i,j}(\log(P_{i,j}/Q_{i,j}) - 1).$$

Note that $\hat{\pi}_n$ has marginals $\hat{a}_n = \frac{1}{n} \sum_{i=1}^n \delta_{x_i}$ and $\hat{b}_n = \frac{1}{n} \sum_{i=1}^n \delta_{y_i}$. We can interpret $\hat{\pi}_n$ as the matrix $\hat{P}_n = \frac{1}{n} \mathrm{Id}_{n \times n}$ and the "noisy" primal problem can be equivalently written as

$$\min_{A \in \mathbb{R}^s} \sup_{P \in \mathbb{R}_+^{n \times n}} \lambda \|A\|_1 + \langle \Phi_n A, P - \hat{P}_n \rangle - \frac{\varepsilon}{2} H(P) \quad s.t. \quad P\mathbb{1} = \frac{1}{n}\mathbb{1} \quad \text{and} \quad P^\top \mathbb{1} = \frac{1}{n}\mathbb{1}, \qquad (\mathscr{P}_n)$$

where we write $H(P) \triangleq H(P | \frac{1}{n^2} \mathbb{1} \otimes \mathbb{1})$ and $\Phi_n A = \sum_k A_k C_k$, where $C_k = (\mathbf{C}_k(x_i, y_j))_{i,j \in [n]} \in \mathbb{R}^{n \times n}$. Note that the finite-dimensional problem has the same invariances as (iOT$-\ell_1(\hat{\pi})$), so, we will take $C_k$ to be centred so that for all $i$, $\sum_i (C_k)_{i,j} = 0$ and for all $j$, $\sum_j (C_k)_{i,j} = 0$. The finite sample Kantorovich formulation is

$$\inf_{A,F,G \in \mathscr{S}_n} \mathscr{K}_n(A, F, G) \quad \text{where} \quad \mathscr{K}_n(A, F, G) \triangleq \mathscr{J}_n(A, F, G) + \lambda \|A\|_1 \quad \text{and} \qquad (\mathscr{K}_n)$$

$$\mathscr{J}_n(A, F, G) \triangleq -\sum_{i,j} \left( F_i \oplus G_j + (\Phi_n A)_{i,j} \right) \left( \hat{P}_n \right)_{i,j} + \frac{\varepsilon}{2n^2} \sum_{i,j} \exp\left( \frac{2}{\varepsilon}(F_i + G_j + (\Phi_n A)_{i,j}) \right),$$

and we restrict the optimization of $(\mathscr{K}_n)$ over $\mathscr{S}_n \triangleq \left\{ (A, F, G) \in \mathbb{R}^s \times \mathbb{R}^n \times \mathbb{R}^n ; \sum_j G_j = 0 \right\}$.

## C PROOFS FOR SECTION 3

### C.1 PROOF OF PROPOSITION 2 (DIFFERENTIABILITY OF $W$)

For the strict convexity of $W(A)$ see Lemma 3 in Dupuy & Galichon (2014). The gradient formula can also be found in Dupuy & Galichon (2014) and trivially follows from the envelope theorem

because the optimization problem $W(A)$ has a unique solution. We will only give a proof of the Hessian formula.

The formula for the Hessian follows from the formula for the gradient provided we show that the density $\pi_A$ is continuously differentiable with respect to $A$. The fact that we can swap the order of the operator $\Phi^*$ and partial differentiation follows from the conditions for differentiability under the integral sign which holds since the measures are compactly supported. Without loss of generality, let $\varepsilon = 1$. Then, the optimizer in $W(A)$ is of the form Santambrogio (2015):

$$\pi_A(x, y) = \exp\Big(u_A(x) + v_A(y) + c_A(x, y)\Big),$$

where $u_A(\cdot)$ and $v_A(\cdot)$ satisfy

$$u_A(x) = -\log \int_{\mathscr{Y}} \exp\Big(v_A(y) + c_A(x, y)\Big) d\beta(y), \quad \alpha\text{-a.s.}$$

$$v_A(y) = -\log \int_{\mathscr{X}} \exp\Big(u_A(x) + c_A(x, y)\Big) d\alpha(x), \quad \beta\text{-a.s.}$$

It is known (see *e.g.* Nutz (2021)) that the functions $u_A(\cdot)$ and $v_A(\cdot)$ inherit the modulus of continuity of the cost (in this case the map $(x, y) \rightarrow c_A(x, y))$) and, hence, since we are assuming the measures to be compactly supported, it follows that $u_A \in L^2(\alpha)$ and $v_A \in L^2(\beta)$. Moreover, if $(u_A, v_A)$ solve these two equations then, for any constant $c$, the pair $(u_A + c, v_A - c)$ is also a solution and, hence, to eliminate this ambiguity we consider solutions in $\mathscr{H} = L^2(\alpha) \times L_0^2(\beta)$, where $L_0^2(\beta) = \big\{g \in L^2(\beta) \, ; \, \int g \, d\beta(y) = 0\big\}$.

To show that $W$ is twice differentiable, since $\nabla W(A) = \Phi^* \pi_A$, it is sufficient to show that $A \mapsto u_A$ and $A \mapsto v_A$ are differentiable. To this end, we will apply the Implicit Function Theorem (in Banach spaces) to the map

$$F : \mathscr{H} \times \mathbb{R}^s \rightarrow L^2(\alpha) \times L^2(\beta)$$

$$\begin{bmatrix} u \\ v \\ A \end{bmatrix} \rightarrow \begin{bmatrix} u + \log \int_{\mathscr{Y}} \exp\Big(v(y) + c_A(x, y)\Big) d\beta(y) \\ v + \log \int_{\mathscr{X}} \exp\Big(u(x) + c_A(x, y)\Big) d\alpha(x) \end{bmatrix},$$

since we have $F(u_A, v_A, A) = 0$. The partial derivative of $F$ at $(u_A, v_A, A)$, denoted by $\partial_{u,v} F(u_A, v_A, A)$, is the linear map defined by

$$\Big(\partial_{u,v} F(u_A, v_A, A)\Big)(f, g) = \Big(f + \int_{\mathscr{Y}} p_A(\cdot, y) g(y) \, d\beta(y), g + \int_{\mathscr{X}} q_A(x, \cdot) f(x) \, d\alpha(x)\Big), \tag{12}$$

where

$$p_A(x, y) = \frac{\exp\Big(v_A(y) + c_A(x, y)\Big)}{\int_{\mathscr{Y}} \exp\Big(v_A(y) + c_A(x, y)\Big) d\beta(y)} \quad \text{and} \quad q_A(x, y) = \frac{\exp\Big(u_A(x) + c_A(x, y)\Big)}{\int_{\mathscr{X}} \exp\Big(u_A(y) + c_A(x, y)\Big) d\beta(x)}.$$

Note that, since $F(u_A, v_A, A) = 0$, $p_A(x, y) = q_A(x, y) = \pi_A(x, y)$. Moreover, since $\pi_A$ has marginals $\alpha$ and $\beta$, it follows that

$$\Big\langle (f, g), \Big(\partial_{u,v} F(u_A, v_A, A)\Big)(f, g) \Big\rangle$$

$$= \int_{\mathscr{X}} f(x)^2 \, d\alpha(x) + \int_{\mathscr{X} \times \mathscr{Y}} 2 f(x) g(y) \pi_A(x, y) \, d(\alpha \otimes \beta)(x, y) + \int_{\mathscr{Y}} g(y)^2 \, d\beta(y). \tag{13}$$

$$= \int_{\mathscr{X} \times \mathscr{Y}} \Big(f(x) + g(y)\Big)^2 \pi_A(x, y) \, d(\alpha \otimes \beta)(x, y)$$

This shows that $\partial_{u,v} F(u_A, v_A, A)$ is invertible – the last line of 13 is zero if and only if $f \oplus g \equiv 0$ and since $g \in L_0^2(\beta)$ it follows that $g = 0$ and $f = 0$.

To conclude the proof we need to show that $\Big(\partial_{u,v} F(u_A, v_A, A)\Big)^{-1}$ is a bounded operator (see *e.g.* Deimling (2010) for the statement of IFT in Banach spaces) and to show this it is enough to show that, for some constant $C$,

$$\Big\|\Big(\partial_{u,v} F(u_A, v_A, A)\Big)(f, g)\Big\| \geq C \|(f, g)\|.$$

This follows from 13 and the fact that there exists a constant $C$ such that $\pi_A(x, y) \geq C$ for all $x$ and $y$ (see *e.g.* Nutz (2021) for the existence of $C$). In fact, from $g \in L_0^2(\beta)$, we obtain $\int (f(x) + g(y)))^2 \, d(\alpha \otimes \beta)(x, y) = \|(f, g)\|^2$ and, hence, 13 implies that

$$\left\langle (f, g), \left( \partial_{u,v} F(u_A, v_A, A) \right)(f, g) \right\rangle \geq C \|(f, g)\|^2$$

and from Cauchy-Schwarz applied to the left-hand-side we obtain

$$\left\| \left( \partial_{u,v} F(u_A, v_A, A) \right)(f, g) \right\| \geq C \|(f, g)\|,$$

thus concluding the proof.

## C.2    CONNECTION OF THE CERTIFICATE (C) WITH OPTIMIZATION PROBLEM (iOT$-\ell_1(\hat{\pi})$)

As mentioned in Section 3.1, the vector $z^\lambda$ provides insight into support recovery since $\mathrm{Supp}(A^\lambda) \subseteq \{i \,;\, z_i^\lambda = \pm 1\}$. In this section, we provide the proof to Proposition 4, which shows that as $\lambda$ converges to 0, $z^\lambda$ converges to the solution of a quadratic optimization problem (MNC) that we term the *minimal norm certificate.*

Note that the connection with (C) can now be established by noting that if the minimal norm certificate is non-degenerate then the inequality constraints (which correspond to the complement of the support of $\widehat{A}$) in (MNC) can be dropped since they are inactive; in this case (MNC) reduces to a quadratic optimization problem with only equality constraints and whose solution can be seen to be (C) which will thus be non-degenerate as well. The converse is clear. So, under non-degeneracy, (C) can be seen as the *limit* optimality vector and determines the support of $A^\lambda$ when $\lambda$ is small.

To prove Proposition 4, we first show that the vector $z^\lambda$ coincides with the solution to a dual problem of (iOT$-\ell_1(\hat{\pi})$).

**Proposition 10.** *Let $W^*$ be the convex conjugate of $W$. Problem (iOT$-\ell_1(\hat{\pi})$) admits a dual given by*

$$\underset{z}{\arg\min}\, W^*(\hat{\Sigma}_{xy} - \lambda z) \quad \text{subject to} \quad \|z\|_\infty \leq 1, \tag{14}$$

*where $\hat{\Sigma}_{xy} = \Phi^* \hat{\pi}$. Moreover, a pair of primal-dual solutions $(A^\lambda, z^\lambda)$ is related by*

$$z^\lambda = -\frac{1}{\lambda} \nabla_A \mathcal{L}(A^\lambda, \hat{\pi}) \quad \text{and} \quad z^\lambda \in \partial \|A^\lambda\|_1. \tag{15}$$

*Proof.* Observe that we can write $\mathcal{L}(A, \hat{\pi})$ as

$$\mathcal{L}(A, \hat{\pi}) = W(A) - \int_{\mathcal{X} \times \mathcal{Y}} (\Phi A)(x, y) \, d\hat{\pi} = W(A) - \langle \Phi^* \hat{\pi}, A \rangle_F = W(A) - \langle A, \hat{\Sigma}_{xy} \rangle.$$

The *Fenchel Duality Theorem* (see *e.g.* Borwein & Lewis (2006)) yields a dual of (iOT$-\ell_1(\hat{\pi})$) given by

$$\underset{w}{\arg\min}\, W^*(\hat{\Sigma}_{xy} - w) + (\lambda \|\cdot\|_1)^*(w).$$

To conclude the proof just note that the Fenchel conjugate of $\lambda \|\cdot\|_1$ is the indicator of the set $\{v : \|v\|_\infty \leq \lambda\}$ and make a change of variable $z \triangleq 1/\lambda \, w$ to obtain 14. The relationship between any primal-dual pair in Fenchel Duality can also be found in Borwein & Lewis (2006).

$\square$

*Proof of Proposition 4.* We begin by noting that $W^*$ is of class $C^2$ in a neighborhood of $\Sigma_{xy}$ and that

$$\nabla^2 W^*(\hat{\Sigma}_{xy}) = \left( \nabla^2 W(\widehat{A}) \right)^{-1}. \tag{16}$$

To see this, note that Proposition 2 together with the assumption on $\hat{\pi}$ implies that

$$\hat{\Sigma}_{xy} = \nabla W(\widehat{A}).$$

Moreover, since $W(A)$ is twice continuously differentiable and strictly convex (see Proposition 2), it follows (see *e.g.* Corollary 4.2.9 in Hiriart-Urruty & Lemaréchal (1993)) that $W^*(\cdot)$ is $C^2$ and strictly convex in a neighborhood of $\hat{\Sigma}_{xy}$ and that (16) holds.

Now observe that, since $\nabla W^*(\hat{\Sigma}_{xy}) = \nabla W^*\big(\nabla W(\widehat{A})\big) = \widehat{A}$, we can rewrite $\partial \big\| \widehat{A} \big\|_1$ as

$$\partial \big\| \widehat{A} \big\|_1 = \underset{z}{\arg\min} \big\langle -z, \nabla W^*(\hat{\Sigma}_{xy}) \big\rangle \quad \text{subject to} \quad \|z\|_\infty \leqslant 1 \tag{17}$$

Observe that since $z^{ambda}$ are uniformly bounded vectors due to the constraint set in 14, there is a convergent subsequence converging to some $z^*$. We later deduce that all limit points are the same and hence, the full sequence $z^\lambda$ converges to $z^*$. Let $\lambda_n$ be such that $\lim_{\lambda_n \to 0} z^{\lambda_n} = z^*$, and let $z^0$ be any element in $\partial \big\| \widehat{A} \big\|_1$. We have that

$$\big\langle -z^0, \nabla W^*(\hat{\Sigma}_{xy}) \big\rangle \leqslant \big\langle -z^{\lambda_n}, \nabla W^*(\hat{\Sigma}_{xy}) \big\rangle \leqslant \frac{1}{\lambda_n} \Big( W^*\big(\hat{\Sigma}_{xy} - \lambda_n z^{\lambda_n}\big) - W(\hat{\Sigma}_{xy}) \Big)$$

$$\leqslant \frac{1}{\lambda_n} \Big( W^*\big(\hat{\Sigma}_{xy} - \lambda_n z^0\big) - W(\hat{\Sigma}_{xy}) \Big),$$

where the first inequality is the optimality of $z^0$, the second inequality is the gradient inequality of convex functions and the last inequality follows from the optimality of $z^\lambda$. Taking the limit as $\lambda_n \to 0$ we obtain that

$$\big\langle -z^0, \nabla W^*(\hat{\Sigma}_{xy}) \big\rangle = \big\langle -z^*, \nabla W^*(\hat{\Sigma}_{xy}) \big\rangle,$$

showing that $z^* \in \partial \big\| \widehat{A} \big\|_1$. We now finish the proof by showing that

$$\big\langle z^*, \nabla^2 W^*(\hat{\Sigma}_{xy}) z^* \big\rangle \leqslant \big\langle z^0, \nabla^2 W^*(\hat{\Sigma}_{xy}) z^0 \big\rangle. \tag{18}$$

Since $W^*(\cdot)$ is $C^2$ in a neighborhood of $\hat{\Sigma}_{xy}$, Taylor's theorem ensures that there exists a remainder function $R(x)$ with $\lim_{x \to \hat{\Sigma}_{xy}} R(x) = 0$ such that

$$\langle -z^{\lambda_n}, \nabla W^*(\hat{\Sigma}_{xy}) \rangle + \frac{\lambda_n}{2} \big\langle z^{\lambda_n}, \nabla^2 W^*(\hat{\Sigma}_{xy}) z^{\lambda_n} \big\rangle + R(\hat{\Sigma}_{xy} + \lambda_n z^{\lambda_n}) \lambda_n^2$$

$$= \frac{1}{\lambda_n} \Big( W^*\big(\hat{\Sigma}_{xy} - \lambda_n z^{\lambda_n}\big) - W^*\big(\hat{\Sigma}_{xy}\big) \Big) \leqslant \frac{1}{\lambda_n} \Big( W^*\big(\hat{\Sigma}_{xy} - \lambda_n z^0\big) - W^*\big(\hat{\Sigma}_{xy}\big) \Big)$$

$$= \langle -z^0, \nabla W^*(\hat{\Sigma}_{xy}) \rangle + \frac{\lambda_n}{2} \big\langle z^0, \nabla^2 W^*(\hat{\Sigma}_{xy}) z^0 \big\rangle + R(\hat{\Sigma}_{xy} + \lambda_n z^0) \lambda_n^2$$

$$\leqslant \langle -z^{\lambda_n}, \nabla W^*(\hat{\Sigma}_{xy}) \rangle + \frac{\lambda_n}{2} \big\langle z^0, \nabla^2 W^*(\hat{\Sigma}_{xy}) z^0 \big\rangle + R(\hat{\Sigma}_{xy} + \lambda_n z^0) \lambda_n^2,$$

where we used the optimality of $z^0$ and of $z^{\lambda_n}$. We conclude that

$$\frac{1}{2} \big\langle z^{\lambda_n}, \nabla^2 W^*(\hat{\Sigma}_{xy}) z^{\lambda_n} \big\rangle + R(\hat{\Sigma}_{xy} + \lambda_n z^{\lambda_n}) \lambda_n \leqslant \frac{1}{2} \big\langle z^0, \nabla^2 W^*(\hat{\Sigma}_{xy}) z^0 \big\rangle + R(\hat{\Sigma}_{xy} + \lambda_n z^0) \lambda_n.$$

Taking the limit establishes 18. Since $z^0$ was an arbitrary element in $\partial \big\| \widehat{A} \big\|_1$, we obtain that the limit of $z^{\lambda_n}$ is

$$z^* = \underset{z}{\arg\min} \big\langle z, \big(\nabla^2 W(\widehat{A})\big)^{-1} z \big\rangle \quad \text{subject to} \quad z \in \partial \big\| \widehat{A} \big\|_1$$

where we used 16. Finally, observe that $z^*$ was an arbitrary limit point of $z^\lambda$ and we showed that all limit points are the same; this is enough to conclude the result. $\qquad\square$

## D   PROOF OF PROPOSITION 6

The proof of this statement relies on strong convexity of $\mathscr{I}_n$. Similar results have been proven in the context of entropic optimal transport (e.g. Genevay et al. (2019); Mena & Niles-Weed (2019)). Our proof is similar to the approach taken in Rigollet & Stromme (2022).

Let $(A_\infty, f_\infty, g_\infty)$ minimize $(\mathscr{K}_\infty)$. Note that $p_\infty \alpha \otimes \beta$ with

$$p_\infty(x, y) = \exp\left(\frac{2}{\varepsilon}\left(\Phi A_\infty(x, y) + f_\infty(x) + g_\infty(y)\right)\right)$$

minimizes $(\text{iOT} - \ell_1(\hat{\pi}))$. Let

$$P_\infty = \frac{1}{n^2}(p_\infty(x_i, y_j))_{i,j}, \quad F_\infty = (f_\infty(x_i))_i \quad \text{and} \quad G_\infty = (g_\infty(y_j))_j.$$

Note that by optimality of $A_\infty$, $\|A_\infty\|_1 \leq \|\hat{A}\|_1$ – this can be seen by comparing the objective $(\text{iOT} - \ell_1(\hat{\pi}))$ at $A_\infty$ and $\hat{A}$. Moreover, due to the uniform bounds on $f_\infty, g_\infty$ from Lemma 12, $p_\infty$ is uniformly bounded away from 0 by $\exp(-C/\lambda)$ for some constant $C$ that depends on $\hat{\pi}$.

Let $P_n$ minimise $(\mathscr{P}_n)$, we know it is of the form

$$P_n = \frac{1}{n^2}\exp\left(\frac{2}{\varepsilon}(\Phi_n A_n + F_n \oplus G_n)\right).$$

for vectors $A_n, F_n, G_n$.

The 'certificates' are $\Phi^* \varphi_\infty$ and $z_n \triangleq \Phi_n^* \varphi_n$ where

$$\varphi_\infty = \frac{1}{\lambda}\left(p_\infty \alpha \otimes \beta - \hat{\pi}\right) \quad \text{and} \quad \varphi_n = \frac{1}{\lambda}\left(P_n - \hat{P}_n\right).$$

Note that $z_n = \Phi^* \varphi_\infty = -\frac{1}{\lambda}\nabla_A \mathscr{L}(A^\infty, \hat{\pi})$. The goal is to bound $\|\Phi^* \varphi_\infty - \Phi_n^* \varphi_n\|_\infty$ so that nondegeneracy of $z_\infty$ would imply nondegeneracy of $\Phi_n^* \varphi_n$. Note that $\Phi_n^* P_n = \Phi^* \hat{\pi}_n$. So, by the triangle inequality,

$$\|\Phi^* \varphi_\infty - \Phi_n^* \varphi_n\|_\infty \leq \frac{1}{\lambda}\|\Phi^*\left(p_\infty \alpha \otimes \beta\right) - \Phi_n^* P_n\|_\infty + \frac{1}{\lambda}\|\Phi^*\left(\hat{\pi}_n - \hat{\pi}\right)\|_\infty$$

$$\leq \frac{1}{\lambda}\|\Phi_n^* P_\infty - \Phi_n^* P_n\|_\infty + \frac{1}{\lambda}\|\Phi^*\left(p_\infty \alpha \otimes \beta\right) - \Phi_n^* P_\infty\|_\infty + \frac{1}{\lambda}\|\Phi^*\left(\hat{\pi}_n - \hat{\pi}\right)\|_\infty$$

The last two terms on the RHS can be controlled using Proposition 20, and are bounded by $\mathcal{O}(tn^{-1/2})$ with probability at least $1 - \mathcal{O}(\exp(-t^2))$ for $t > 0$. For the first term on the RHS, letting $Z = P_\infty - P_n$,

$$\|\Phi_n^* Z\|_\infty = \left\|\sum_{i,j=1}^n \mathbf{C}(x_i, y_j) Z_{i,j}\right\|_\infty$$

$$\leq \|\mathbf{C}\|_\infty \sqrt{\frac{1}{n^2}\sum_{i,j}\left(\exp\left(\frac{2}{\varepsilon}(\Phi_n A_n + F_n \oplus G_n)\right) - \exp\left(\frac{2}{\varepsilon}(\Phi_n A_\infty + F_\infty \oplus G_\infty)\right)\right)_{i,j}^2}$$

Let $L \triangleq 2(\|\Phi_n A_n + F_n \oplus G_n)\|_\infty \vee \|\Phi_n A_\infty + F_\infty \oplus G_\infty)\|_\infty$. According to the Lipschitz continuity of the exponential

$$\frac{1}{n^2}\sum_{i,j}\left(\exp\left(\frac{2}{\varepsilon}(\Phi_n A_n + F_n \oplus G_n)\right) - \exp\left(\frac{2}{\varepsilon}(\Phi_n A_\infty + F_\infty \oplus G_\infty)\right)\right)_{i,j}^2 \tag{19}$$

$$\leq \frac{4\exp(L/\varepsilon)}{\varepsilon^2 n^2}\sum_{i,j}\left((\Phi_n A_n + F_n \oplus G_n) - (\Phi_n A_\infty + F_\infty \oplus G_\infty)\right)_{i,j}^2 \tag{20}$$

$$\leq \frac{12\exp(L/\varepsilon)}{\varepsilon^2}\left(\frac{1}{n^2}\sum_{i,j}(\Phi_n A_n - \Phi_n A_\infty)_{i,j}^2 + \frac{1}{n}\sum_i(F_n - F_\infty)_i^2 + \frac{1}{n}\sum_j(G_n - G_\infty)_j^2\right) \tag{21}$$

By strong convexity properties of $\mathscr{J}_n$ and hence $\mathscr{K}_n$, it can be shown (see Prop 14) that (21) is upper bounded up to a constant by

$$\varepsilon^{-1}\exp(L/\varepsilon)\left(\mathscr{K}_n(F_\infty, G_\infty, A_\infty) - \mathscr{K}_n(F_n, G_n, A_n)\right)$$

$$\leq \frac{\exp(L/\varepsilon)}{4}\left(\left\|n^{-2}(\Phi_n^* \Phi_n)^{-1}\right\|\left\|(\partial_A \mathscr{J}_n(A_\infty, F_\infty, G_\infty) + \lambda \xi_\infty)\right\|^2\right.$$

$$+ n \left\| \partial_F \mathscr{J}_n(A_\infty, F_\infty, G_\infty) \right\|^2 + n \left\| \partial_G \mathscr{J}_n(A_\infty, F_\infty, G_\infty) \right\|^2 \Big)$$

where $\xi_\infty = \frac{1}{\lambda}(\Phi^* \hat{\pi} - \Phi^*(p_\infty \alpha \otimes \beta)) \in \partial \| A_\infty \|_1$. By Lemma 11 and Lemma 12, $L = \mathscr{O}(\| \widehat{A} \|_1)$ with probability at least $1 - \mathscr{O}(\exp(-t^2))$ if $\lambda \gtrsim t n^{-\frac{1}{2}}$. Finally,

$$\left\| (\partial_A \mathscr{J}_n(A_\infty, F_\infty, G_\infty) + \lambda \xi_\infty \right\|^2 = \left\| \Phi_n^* \hat{P} + \Phi_n^* P_\infty + \lambda \xi_\infty \right\|^2$$
$$\leqslant 2 \left( \left\| \Phi_n^* \hat{P} - \Phi^* \hat{\pi} \right\|^2 + \left\| \Phi_n^* P_\infty - \Phi^*(p_\infty \alpha \otimes \beta) \right\|^2 \right)$$

$$n \left\| \partial_F \mathscr{J}_n(A_\infty, F_\infty, G_\infty) \right\|_2^2 = \frac{1}{n} \sum_{i=1}^n \left( 1 - \frac{1}{n} \sum_{j=1}^n p_\infty(x_i, y_j) \right)^2$$

$$n \left\| \partial_G \mathscr{J}_n(A_\infty, F_\infty, G_\infty) \right\|_2^2 \Big) = \frac{1}{n} \sum_{j=1}^n \left( 1 - \frac{1}{n} \sum_{i=1}^n p_\infty(x_i, y_j) \right)^2 \Big).$$

We show in Propositions 16, 19 and 20 that these are bounded by $\mathscr{O}(t^2 n^{-1})$ with probability at least $1 - \mathscr{O}(\exp(-t^2))$ and from Proposition 23, assuming $n \gtrsim \log(2s)$, $\left\| (n^{-2} \Phi_n^* \Phi_n)^{-1} \right\| \lesssim 1$ with probability at least $1 - \mathscr{O}(\exp(-n))$.

So, for some constant $C > 0$, $\lambda \left\| \Phi^* \varphi_\infty - \Phi_n^* \varphi_n \right\|_\infty \lesssim \exp(C \| \widehat{A} \|_1 / \varepsilon) \frac{t}{\sqrt{n}}$ with probability at least $1 - \exp(-t^2)$. The second statement follows by combining our bound for (21) with the fact that $\left\| (n^{-2} \Phi_n^* \Phi_n)^{-1} \right\| \lesssim 1$.

In the following subsections, we complete the proof by establishing the required strong convexity properties of $\mathscr{J}_n$ and bound $\nabla \mathscr{J}_n$ at $(A_\infty, F_\infty, G_\infty)$ using concentration inequalities. The proofs to some of these results are verbatim to the results of Rigollet & Stromme (2022) for deriving sampling complexity bounds in eOT, although they are included due to the difference in our setup.

## D.1  STRONG CONVEXITY PROPERTIES OF $\mathscr{J}_n$

In this section, we present some of the key properties of $\mathscr{J}_n$. Recall that since we assume that $\alpha, \beta$ are compactly supported, up to a rescaling of the space, we assume without loss of generality that for all $k$, $|\mathbf{C}_k(x, y)| \leqslant 1$.

**Lemma 11.** *Let $A_n$ minimize (iOT$-\ell_1(\hat{\pi}_n)$). Assume that for some constant $C_1 > 0$,*

$$t \exp(C_1/\varepsilon) n^{-\frac{1}{2}} \lesssim \lambda \min\left(1, \| \widehat{A} \|_1\right),$$

*then with probability at least $1 - \mathscr{O}(\exp(-t^2))$,*

$$\| A_n \|_1 \leqslant 2 \| \widehat{A} \|$$

*Proof.* Let $A$ be a minimizer to (iOT$-\ell_1(\hat{\pi}_n)$). By optimality of $A$, $\lambda \| A \| + \mathscr{L}(A, \hat{\pi}_n) \leqslant \lambda \| \widehat{A} \|_1 + \mathscr{L}(\widehat{A}, \hat{\pi}_n)$. Writing $\hat{P}_n = \frac{1}{n} \mathrm{Id}$ and $\Phi_n A = \left( \sum_{k=1}^s A_k \mathbf{C}_k(x_i, y_j) \right)_{i,j=1}^n$,

$$\lambda \| A \|_1 - \langle \hat{P}_n, \Phi_n A \rangle + \langle P, \Phi_n A \rangle - \frac{\varepsilon}{2} H(P) \leqslant \lambda \| \widehat{A} \|_1 + \mathscr{L}(\widehat{A}, \hat{\pi}_n) \tag{22}$$

for all $P$ satisfying the marginal constraints $P \mathbb{1} = \frac{1}{n} \mathbb{1}$ and $P^\top \mathbb{1} = \frac{1}{n} \mathbb{1}$.

Note that since $\mathscr{L}(\widehat{A}, \hat{\pi}) + \mathrm{KL}(\hat{\pi} | \alpha \otimes \beta) = 0$,

$$\mathscr{L}(\widehat{A}, \hat{\pi}_n) = \mathscr{L}(\widehat{A}, \hat{\pi}_n) - \mathscr{L}(\widehat{A}, \hat{\pi}) - \mathrm{KL}(\hat{\pi} | \alpha \otimes \beta) \tag{23}$$
$$= -\langle \Phi \widehat{A}, \hat{\pi} - \hat{\pi}_n \rangle + W_{\hat{\pi}_n}(\widehat{A}) - W_{\hat{\pi}}(\widehat{A}) - \mathrm{KL}(\hat{\pi} | \alpha \otimes \beta). \tag{24}$$

The first two terms can be shown to be $\mathscr{O}(n^{-\frac{1}{2}})$: Indeed,

$$\langle \Phi \widehat{A}, \hat{\pi} - \hat{\pi}_n \rangle = \frac{1}{n} \sum_{i=1}^n Z_i, \quad \text{where} \quad Z_i \triangleq \left( \langle \widehat{A}, \mathbf{C}(x_i, y_i) \rangle - \mathbb{E}[\langle \widehat{A}, \mathbf{C}(x_i, y_i) \rangle] \right)$$

is the sum of $n$ terms with mean zero. Moreover, $|Z_i| \leqslant 2 \| \widehat{A} \|$. By Lemma 18, $\langle \Phi \widehat{A}, \hat{\pi} - \hat{\pi}_n \rangle \leqslant \frac{8 \| \widehat{A} \|^2 t}{\sqrt{n}}$ with probability at least $1 - 2 \exp(-t^2)$.

The bound

$$W_{\hat{\pi}_n}(\widehat{A}) - W_{\hat{\pi}}(\widehat{A}) = \mathcal{O}(\exp(-C/\varepsilon)\, t n^{-\frac{1}{2}})$$

with probability $1 - \mathcal{O}(\exp(t^2))$ is a due to the sample complexity of eOT Rigollet & Stromme (2022).

Plugging this back into (22), we obtain

$$\lambda \|A\|_1 \leq \lambda \|\widehat{A}\|_1 + \mathcal{O}(n^{-\frac{1}{2}}) + \frac{\varepsilon}{2} \underbrace{(H(P) - \mathrm{KL}(\hat{\pi}|\alpha \otimes \beta))}_{T_1} + \underbrace{(\langle \hat{P}_n, \Phi_n A \rangle + \langle P, \Phi_n A \rangle)}_{T_2} \qquad (25)$$

It remains to bound the terms $T_1$ and $T_2$. Intuitively, if $\hat{\pi} = \hat{p}\alpha \otimes \beta$ has density $\hat{p}$, then we can show that $T_1, T_2 = \mathcal{O}(n^{-\frac{1}{2}})$ by choosing $P = Q_n \triangleq \frac{1}{n^2}(\hat{p}(x_i, y_j))_{i,j=1}^n$. However, $Q_n$ will only approximately satisfy the marginal constraints $Q_n \mathbb{1} \approx \frac{1}{n}\mathbb{1}$ and $Q_n^\top \mathbb{1} \approx \frac{1}{n}\mathbb{1}$ (this approximation can be made precise using Proposition 16). So, we insert into the above inequality $P = \tilde{Q}_n$ with $\tilde{Q}_n$ being the projection of $Q_n$ onto the constraint set $\mathcal{U}(\frac{1}{n}\mathbb{1}_n, \frac{1}{n}\mathbb{1}_n)$.

By (Altschuler et al., 2017, Lemma 7), the projection $\tilde{Q}_n$ satisfies

$$\left\|\tilde{Q}_n - Q_n\right\|_1 \leq 2 \left\|Q_n\mathbb{1} - \frac{1}{n}\mathbb{1}\right\|_1 + 2 \left\|Q_n^\top\mathbb{1} - \frac{1}{n}\mathbb{1}\right\|_1. \qquad (26)$$

By Proposition 16, with probability at least $1 - \mathcal{O}(\exp(-t^2))$,

$$\left\|Q_n\mathbb{1} - \frac{1}{n}\mathbb{1}\right\|_1 = \frac{1}{n}\sum_{i=1}^n \left|\frac{1}{n}\sum_{j=1}^n \hat{p}(x_i, y_j) - 1\right| \leq \sqrt{\frac{1}{n}\sum_{i=1}^n \left|\frac{1}{n}\sum_{j=1}^n \hat{p}(x_i, y_j) - 1\right|^2} = \mathcal{O}\left(\frac{t}{\sqrt{n}}\right).$$

Similarly, $\left\|Q_n\mathbb{1} - \frac{1}{n}\mathbb{1}\right\|_1 = \mathcal{O}(n^{-\frac{1}{2}})$ with high probability.

Note that

$$\left|\langle \tilde{Q}_n - Q_n, \Phi_n A \rangle\right| = \left|\sum_{k=1}^s \sum_{i,j=1}^n A_k (C_k)_{i,j}(\tilde{Q}_n - Q_n)_{i,j}\right| \qquad (27)$$

$$\leq \max_k \|C_k\|_\infty \|A\|_1 \left\|Q_n - \tilde{Q}_n\right\|_1 \qquad (28)$$

Moreover,

$$\langle \hat{P}_n - Q_n, \Phi_n A \rangle \leq \|A\| \left\|\Phi_n^*(\hat{P}_n - Q_n)\right\| \qquad (29)$$

By Proposition 19 and 20, with probability at least $1 - \mathcal{O}(\exp(-t^2))$,

$$\mathbb{E}\left\|\Phi_n^*\hat{P}_n - \Phi^*\hat{\pi}\right\| = \mathcal{O}(t n^{-\frac{1}{2}}) \quad \text{and} \quad \mathbb{E}\left\|\Phi_n^* Q_n - \Phi^*\hat{\pi}\right\| = \mathcal{O}(t n^{-\frac{1}{2}}).$$

So, we have $T_2 \leq C n^{-\frac{1}{2}} \|A\|_1$.

We now consider the term $T_1$ in (25). Since $\hat{p}$ is uniformly bounded from above by $\exp(C/\varepsilon)$ and from below with constant $\exp(-C/\varepsilon)$ for some $C > 0$, one can check from the projection procedure of Altschuler et al. (2017) that $\tilde{Q}_n$ is also uniformly bounded from above by $\frac{1}{n^2}\exp(C/\varepsilon)$ and away from zero by $\frac{1}{n^2}\exp(-C/\varepsilon)$ for some $C > 0$. So,

$$|T_1| \leq \left|H(\tilde{Q}_n) - H(Q_n)\right| + \left|H(Q_n) - \mathrm{KL}(\hat{\pi}|\alpha \otimes \beta)\right| \qquad (30)$$

$$\lesssim e^{C/\varepsilon}\left\|\tilde{Q}_n - Q_n\right\|_1 + \left|H(Q_n) - \mathrm{KL}(\hat{\pi}|\alpha \otimes \beta)\right| \qquad (31)$$

We can use the bound (26) to see that $\left\|\tilde{Q}_n - Q_n\right\|_1 \lesssim t/\sqrt{n}$ with probability at least $1 - \exp(-t^2)$. To bound $T_3 \triangleq H(Q_n) - \mathrm{KL}(\hat{\pi}|\alpha \otimes \beta)$, note that Moreover,

$$\mathbb{E}[T_3] = \mathbb{E}\left(\frac{1}{n^2}\sum_{i=1}^n \sum_{j=1}^n \log(\hat{p}(x_i, y_j))\hat{p}(x_i, y_j)\right) - \int \log(\hat{p}(x, y))\hat{p}(x, y)\, d\alpha(x)d\beta(y)$$

$$= \left(\frac{n(n-1)}{n^2} - 1\right)\int \log(\hat{p}(x, y))\hat{p}(x, y)\, d\alpha(x)d\beta(y) + \frac{1}{n^2}\sum_{i=1}^n \mathbb{E}\left(\log(\hat{p}(x_i, y_i))\hat{p}(x_i, y_i)\right)$$

$$\leq \frac{-1}{n^2} \int \log(\hat{p}(x,y))\hat{p}(x,y)\,d\alpha(x)\,d\beta(y) + \frac{1}{n^2} \int \log(\hat{p}(x,y))\hat{p}(x,y)^2\,d\alpha(x)\,d\beta(y)$$

$$\leq \frac{1}{n^2}\left(\|\hat{p}\|_\infty - 1\right) \int \log(\hat{p}(x,y))\,d\hat{\pi}(x,y)$$

Therefore, by the bounded differences lemma 15 with

$$f(z_1,\ldots,z_n) = \frac{1}{n^2} \sum_{i=1}^n \sum_{j=1}^n \log(\hat{p}(x_i,y_j))\hat{p}(x_i,y_j) - \int \log(\hat{p}(x,y))\hat{p}(x,y)\,d\alpha(x)\,d\beta(y)$$

where $z_i = (x_i, y_i)$. Then, letting $z_j = z_j'$ for all $j \neq i$, the bounded differences property is satisfied with

$$f(z_1,\ldots,z_n) - f(z_1',\ldots,z_n')$$

$$= \frac{1}{n^2}\left(\sum_{\substack{j=1 \\ j\neq i}}^n + \sum_{j=1}^n\right)\left(\log(\hat{p}(x_i,y_j))\hat{p}(x_i,y_j) - \log(\hat{p}(x_i',y_j))\hat{p}(x_i',y_j)\right)$$

$$\leq \frac{4(\|\hat{p}\|_\infty + 1)^2)}{n} \triangleq c.$$

So, with probability at least $1 - 2\exp\left(-t^2\right)$, $|T_3| \leq \frac{1}{n^2} + \frac{2t(\|\hat{p}\|_\infty + 1)}{\sqrt{n}}$.

In summary, with probability $1 - \mathcal{O}(\exp(-t^2))$,

$$\left(\lambda - C_2 t \exp(C_1/\varepsilon) n^{-\frac{1}{2}}\right)\|A\|_1 \leq \lambda\left\|\widehat{A}\right\|_1 + \frac{C_2(\exp(C_1/\varepsilon)t}{\sqrt{n}}$$

for some constants $C_1, C_2 > 0$. So, choosing

$$C_2 t \exp(C_1/\varepsilon) n^{-\frac{1}{2}} \leq \min\left(\lambda/4, \lambda\left\|\widehat{A}\right\|_1 / 2\right),$$

we have

$$\|A\|_1 \leq 2\left\|\widehat{A}\right\|$$

with probability at least $1 - \exp(-t^2)$.

$\square$

**Lemma 12.** *Let $(A, F, G)$ minimize $(\mathcal{K}_n)$. Let $C_A = \Phi_n A$. Then,*

$$F_i \in [-3\|C_A\|_\infty, \|C_A\|_\infty] \quad and \quad G_i \in [-2\|C_A\|_\infty, 2\|C_A\|_\infty].$$

*Moreover,* $\exp(4\|C_A\|_\infty \varepsilon^{-1}) \geq \exp\left(\frac{2}{\varepsilon}\left(F_i + G_j + (\Phi_n A)_{i,j}\right)\right) \geq \exp(-6\|C_A\|_\infty \varepsilon^{-1})$. *Note that* $\|C_A\|_\infty \leq \|A\|_1$.

*If $(A, f, g)$ minimize $(\mathcal{K}_\infty)$. Let $c_A = \Phi A$. Then, for all $x, y$*

$$f(x) \in [-3\|c_A\|_\infty, \|c_A\|_\infty] \quad and \quad g(y) \in [-2\|c_A\|_\infty, 2\|c_A\|_\infty].$$

*Moreover,* $\exp(4\|c_A\|_\infty \varepsilon^{-1}) \geq \exp\left(\frac{2}{\varepsilon}\left(f \oplus + g + (\Phi A)_{i,j}\right)\right) \geq \exp(-6\|c_A\|_\infty \varepsilon^{-1})$.

*Proof.* This proof is nearly identical to (Rigollet & Stromme, 2022, Prop. 10): Let $A, F, G$ minimize $(\mathcal{K}_n)$.

By the marginal constraints for $P_n \triangleq \frac{1}{n^2}\left(\exp\left(\frac{2}{\varepsilon}(F \oplus G + \Phi_n A)\right)\right)_{i,j}$ given in $(\mathcal{P}_n)$,

$$1 = \frac{1}{n}\sum_j \exp\left(\frac{2}{\varepsilon}(\Phi A(x_i, y_j) + F_i + G_j)\right)$$

$$\geq \exp\left(\frac{2}{\varepsilon}(-\|C_A\|_\infty + F_i)\right)\sum_j \frac{1}{n}\exp\left(2G_j/\varepsilon\right) \geq \exp\left(\frac{2}{\varepsilon}(-\|C_A\|_\infty + F_i)\right)$$

(32)

where we use Jensen's inequality and the assumption that $\sum_j G_j = 0$ for the second. So, $F_i \leq \|C_A\|_\infty$ for all $i \in [n]$. Using the other marginal constraint for $P_n$ along with this bound on $F_i$ implies that

$$1 = \frac{1}{n}\sum_i \exp\left(\frac{2}{\varepsilon}(\Phi A(x_i, y_j) + F_i + G_j)\right) \leq \exp\left(\frac{2}{\varepsilon}\left(2\|C_A\|_\infty + G_j)\right)\right)$$

So, $G_{n,j} \geq -2\|C_A\|_\infty$.

To prove the reverse bounds, we now show that $\sum_i F_i$ is lower bounded: Note that since $H(P) \geq -1$, by duality between $(\mathscr{P}_n)$ and $(\mathscr{K}_n)$,

$$\|C_A\|_\infty + \frac{\varepsilon}{2} \geq \langle \Phi_n A, P\rangle - \frac{\varepsilon}{2}H(P) = -\frac{1}{n}\sum_i F_i - \frac{1}{n}\sum_j G_j + \frac{\varepsilon}{2n^2}\sum_{i,j}\exp\left(\frac{2}{\varepsilon}(F_i + G_j + (\Phi_n A)_{ij})\right)$$

By assumption, $\sum_j G_j = 0$ and $\sum_{i,j}\exp\left(\frac{2}{\varepsilon}(F_i + G_j + (\Phi_n A)_{ij})\right) = n^2$. So,

$$\frac{1}{n}\sum_i F_i \geq -\|C_A\|_\infty.$$

By repeating the argument in (32), we see that

$$1 \geq \exp(2/\varepsilon(-\|C_A\|_\infty + G_j))\exp\left(\frac{2}{n\varepsilon}\sum_i F_i\right) \geq \exp(2/\varepsilon(-2\|C_A\|_\infty + G_j)).$$

So, $G_j \leq 2\|C_A\|_\infty$ and $F_i \geq -3\|C_A\|_\infty$.

The proof for $(\mathscr{K}_\infty)$ is nearly identical and hence omitted. $\qquad\square$

Similarly to (Rigollet & Stromme, 2022, Lemma 11), we derive the following strong convexity bound for $\mathscr{J}_n$:

**Lemma 13.** *The functional $\mathscr{J}_n$ is strongly convex with*

$$\mathscr{J}_n(A', F', G') \geq \mathscr{J}_n(A, F, G) + \langle\nabla\mathscr{J}_n(A, F, G), (A', F', G') - (A, F, G)\rangle$$
$$+ \frac{\exp(-L/\varepsilon)}{\varepsilon}\left(\frac{1}{n^2}\|\Phi_n(A - A')\|_2^2 + \frac{1}{n}\|F - F'\|_2^2 + \frac{1}{n}\|G - G'\|_2^2\right), \qquad (33)$$

*for some $L = \mathcal{O}(\|A\|_1 \vee \|A'\|_1)$.*

*Proof.* To establish the strong convexity inequality, let

$$h(t) \triangleq \mathscr{J}((1-t)A + tA', (1-t)f + tf', (1-t)g + tg').$$

It suffices to find $\delta > 0$ such that for all $t \in [0, 1]$,

$$h''(t) \geq \delta\left(\frac{1}{n^2}\|\Phi_n(A - A')\|_2^2 + \frac{1}{n}\|F - F'\|_2^2 + \frac{1}{n}\|G - G'\|_2^2\right). \qquad (34)$$

Let $Z_t \triangleq ((1-t)F + tF') \oplus ((1-t)G + tG') + ((1-t)\Phi A + t\Phi A')$. Note that

$$h''(t) = \frac{2}{\varepsilon n^2}\sum_{i,j}\exp\left(\frac{2}{\varepsilon}(Z_t)_{i,j}\right)(F'_i - F_i + G'_i - G_i + (\Phi A')_{i,j} - (\Phi A)_{i,j})^2 \qquad (35)$$

Since $\|c_A\|_\infty \vee \|c_{A'}\|_\infty \leq L$, by Lemma 12, $\|Z_t\|_\infty \lesssim \|c_A\|_\infty \vee \|c_{A'}\|_\infty$. So,

$$h''(t) \geq \frac{2}{\varepsilon n^2}\exp(-L/\varepsilon)\sum_{i,j}(F'_i - F_i + G'_i - G_i + (\Phi A')_{i,j} - (\Phi A)_{i,j})^2.$$

By expanding out the brackets and using $\sum_i G_i = 0$ and since $C_k$ are centred ($\sum_i(C_k)_{i,j} = 0$ and $\sum_j(C_k)_{i,j} = 0$), (34) holds with $\delta = \frac{2}{\varepsilon}\exp(-L/\varepsilon)$. $\qquad\square$

Based on this strong convexity result, we have the following bound.

**Proposition 14.** *Let $(A_n, F_n, G_n)$ minimise $\mathscr{F}_n$, then for all $(A, F, G) \in \mathscr{S}$ such that $n^{-2} \exp(F \oplus G + \Phi_n A)$ satisfy the marginal constraints of $(\mathscr{P}_n)$,*

$$\mathscr{K}_n(A, F, G) - \mathscr{K}_n(A_n, F_n, G_n) \geqslant \frac{\exp(-L/\varepsilon)}{\varepsilon} \left( \frac{1}{n^2} \|\Phi_n(A - A_n)\|_2^2 + \frac{1}{n} \|F - F_n\|_2^2 + \frac{1}{n} \|G - G_n'\|_2^2 \right).$$

*and*

$$\mathscr{K}_n(A, F, G) - \mathscr{K}_n(A_n, F_n, G_n) \leqslant \frac{\varepsilon \exp(L/\varepsilon)}{4} \left( \left\| n^{-2}(\Phi_n^* \Phi_n)^{-1} \right\| \left\| (\partial_A \mathscr{J}_n(A, F, G) + \lambda \xi) \right\|^2 \right.$$
$$\left. + n \left\| \partial_F \mathscr{J}_n(A, F, G) \right\|_2^2 + n \left\| \partial_G \mathscr{J}_n(A, F, G) \right\|_2^2 \right).$$

*where $L = \mathcal{O}(\|A\|_1 \vee \|A_n\|_1)$.*

*Proof.* By strong convexity of $\mathscr{J}_n$, we can show that for any $(A, f, g), (A', f', g') \in \mathscr{S}$ and any $\xi \in \partial \|A\|_1$,

$$\mathscr{K}_n(A', F', G') \geqslant \mathscr{K}_n(A, F, G) + \langle \nabla \mathscr{J}_n(A, F, G), (A', F', G') - (A, F, G) \rangle + \lambda \langle \xi, A' - A \rangle$$
$$+ \frac{\delta}{2} \left( \frac{1}{n^2} \|\Phi_n(A - A')\|_2^2 + \frac{1}{n} \|F - F'\|_2^2 + \frac{1}{n} \|G - G'\|_2^2 \right), \tag{36}$$

where $\delta = \frac{2 \exp(-L/\varepsilon)}{\varepsilon}$ with $L = \mathcal{O}(\|A\|_1 \vee \|A'\|_1)$. The first statement follows by letting $(A, F, G) = (A_n, F_n, G_n)$ in the above inequality.

To prove the second statement, let

$$M \triangleq \langle \nabla \mathscr{J}_n(A, F, G), (A', F', G') - (A, F, G) \rangle + \lambda \langle \xi, A' - A \rangle$$
$$+ \frac{\delta}{2} \left( \frac{1}{n^2} \|\Phi_n(A - A')\|_2^2 + \frac{1}{n} \|F - F'\|_2^2 + \frac{1}{n} \|G - G'\|_2^2 \right).$$

By minimising over $(A', F', G')$, note that

$$M \geqslant -\frac{1}{2\delta} \left( n^2 \left\| (\partial_A \mathscr{J}_n(A, F, G) + \lambda \xi) \right\|_{(\Phi_n^* \Phi_n)^{-1}}^2 + n \left\| \partial_F \mathscr{J}_n(A, F, G) \right\|_2^2 + n \left\| \partial_G \mathscr{J}_n(A, F, G) \right\|_2^2 \right),$$

So,

$$-M \leqslant \frac{1}{2\delta} \left( \left\| n^{-2}(\Phi_n^* \Phi_n)^{-1} \right\| \left\| (\partial_A \mathscr{J}_n(A, F, G) + \lambda \xi) \right\|^2 + n \left\| \partial_F \mathscr{J}_n(A, F, G) \right\|_2^2 + n \left\| \partial_G \mathscr{J}_n(A, F, G) \right\|_2^2 \right)$$

Finally, note that $\mathscr{K}_n(A, F, G) - \mathscr{K}_n(A', F', G') \geqslant \mathscr{K}_n(A, F, G) - \mathscr{K}_n(A_n, F_n, G_n)$ by optimality of $A_n, F_n, G_n$. $\quad\square$

### D.2 CONCENTRATION BOUNDS

**Lemma 15** (McDiarmid's inequality). *Let $f : \mathscr{X}^n \to \mathbb{R}$ satisfy the bounded differences property:*

$$\sup_{x_i' \in \mathscr{X}} \left| f(x_1, \ldots, x_{i-1}, x_i, x_{i+1}, \ldots, x_n) - f(x_1, \ldots, x_{i-1}, x_i', x_{i+1}, \ldots, x_n) \right| \leqslant c.$$

*Then, given $X_1, \ldots, X_n$ random variables with $X_i \in \mathscr{X}$, for any $t > 0$,*

$$\mathbb{P}\left( \left| f(X_1, \ldots, X_n) - \mathbb{E}[f(X_1, \ldots, X_n)] \right| \geqslant t \right) \leqslant 2 \exp(-2t^2/(nc^2)).$$

Given random vectors $X$ and $Y$, denote $\text{Cov}(X, Y) = \mathbb{E}\langle Y - \mathbb{E}[Y], X - \mathbb{E}[X] \rangle$ and $\text{Var}(X) = \text{Cov}(X, X)$.

**Proposition 16.** *Let $\pi$ have marginals $\alpha$ and $\beta$ and let $p = \frac{d\pi}{d(\alpha \otimes \beta)}$. Let $(x_i, y_i) \sim \hat{\pi}$ where $\hat{\pi}$ has marginals $\alpha, \beta$. Assume that $\|p\|_\infty \leqslant b$. Then,*

$$\mathbb{E}\left[ \frac{1}{n} \sum_{j=1}^n \left( 1 - \frac{1}{n} \sum_{i=1}^n p(x_i, y_j) \right)^2 \right] \leqslant \frac{(b+1)^2}{n}$$

*and*

$$\frac{1}{n} \sum_{j=1}^n \left( 1 - \frac{1}{n} \sum_{i=1}^n p(x_i, y_j) \right)^2 \leqslant \frac{(t + b + 1)^2}{n}.$$

*with probability at least $1 - \exp(-t^2/(4b^2))$.*

*Remark* 17. The bounds also translate to an $\ell_1$ norm on the marginal errors, since by Cauchy-Schwarz,

$$\frac{1}{n} \sum_{j=1}^{n} \left| 1 - \frac{1}{n} \sum_{i=1}^{n} p(x_i, y_j) \right| \leqslant \sqrt{\sum_{j=1}^{n} \frac{1}{n} \left( 1 - \frac{1}{n} \sum_{i=1}^{n} p(x_i, y_j) \right)^2}.$$

Moreover, by Jensen's inequality $\mathbb{E} \frac{1}{n} \sum_{j=1}^{n} \left| 1 - \frac{1}{n} \sum_{i=1}^{n} p(x_i, y_j) \right| \leqslant \sqrt{\mathbb{E} \sum_{j=1}^{n} \frac{1}{n} \left( 1 - \frac{1}{n} \sum_{i=1}^{n} p(x_i, y_j) \right)^2}.$

*Proof.*

$$\mathbb{E} \left[ \frac{1}{n} \sum_{j=1}^{n} \left( 1 - \frac{1}{n} \sum_{i=1}^{n} p(x_i, y_j) \right)^2 \right] = \frac{1}{n^3} \sum_{ij,k=1}^{n} \mathbb{E} \left( (1 - p(x_i, y_j))(1 - p(x_k, y_j)) \right).$$

For each $j \in [n]$, we have the following cases for $u_j \triangleq \mathbb{E} \left( (1 - p(x_i, y_j))(1 - p(x_k, y_j)) \right)$:

1. $i = k = j$, then $u_j = \mathbb{E}_{(x,y) \sim \hat{\pi}} (p_\infty(x, y) - 1)^2$. There is 1 such term.

2. $i = j$ and $k \neq j$, then $u_j = \mathbb{E}_{(x,y) \sim \hat{\pi}, z \sim \alpha} \left( (1 - p(x, y))(1 - p(z, y)) \right) = 0$. There are $n - 1$ such terms.

3. $i \neq j$ and $k = j$, then $u_j = \mathbb{E}_{(z,y) \sim \hat{\pi}, x \sim \alpha} \left( (1 - p(x, y))(1 - p(z, y)) \right) = 0$. There are $n - 1$ such terms.

4. $i = k$ and $i \neq j$, then $u_j = \mathbb{E}_{x \sim \alpha, y \sim \beta} (1 - p(x, y))^2$ and there are $(n - 1)$ such terms.

5. $i, j, k$ all distinct. Then, $u_j = 0$ and there are $(n - 1)(n - 2)$ such terms.

Therefore,

$$\frac{1}{n^3} \sum_{ij,k=1}^{n} \mathbb{E} \left( (1 - p(x_i, y_j))(1 - p(x_k, y_j)) \right) = \frac{1}{n^2} \mathbb{E}_{(x,y) \sim \hat{\pi}} (p(x, y) - 1)^2 + \frac{n-1}{n^2} \mathbb{E}_{x \sim \alpha, y \sim \beta} (1 - p(x, y))^2$$

Using $|1 - p(x, y)| \leqslant b + 1$ gives the first inequality.

Note also that letting $V = \frac{1}{n} \sum_{i=1}^{n} \left( (1 - p(x_i, y_j)) \right)_j$, $\|V\|^2 = \sum_{j=1}^{n} \left( \frac{1}{n} \sum_{i=1}^{n} (1 - p(x_i, y_j)) \right)^2$. We will apply Lemma 15 to $f(z_1, \ldots, z_n) = \|V\|$. Let $V' = \frac{1}{n} \sum_{i=1}^{n} \left( (1 - p(x_i', y_j')) \right)_j$ where $x_i, y_j = x_j', y_j'$ for $i, j \geqslant 2$. Then, for all vectors $u$ of norm 1,

$$\langle u, V' - V \rangle = \sum_{j=1}^{n} u_j \frac{1}{n} \sum_{i=1}^{n} (p(x_i', y_j') - p(x_i, y_j))$$

$$= u_1 \frac{1}{n} \sum_{i=1}^{n} (p(x_i', y_1') - p(x_i, y_1)) + \sum_{j=2}^{n} u_j \frac{1}{n} (p(x_1', y_j') - p(x_1, y_j))$$

$$\leqslant \frac{2b}{n} \sum_{j=1}^{n} u_j \leqslant \frac{2b}{\sqrt{n}}.$$

So, by the reverse triangle inequality, $\left| \|V\| - \|V'\| \right| \leqslant \frac{2b}{\sqrt{n}}$. It follows that

$$n^{-1/2} \|V\| \leqslant n^{-1/2} t + n^{-1/2} \mathbb{E} \|V\| \leqslant n^{-1/2} t + \sqrt{n^{-1} \mathbb{E} \|V\|^2} \leqslant n^{-1/2} (t + b + 1).$$

with probability at least $1 - \exp(-t^2/(4b^2))$ as required.

$\square$

### D.2.1 Bounds for the Cost Parametrization

In the following two propositions (Prop 20 and 19), we assume that $\Phi_n$ is defined with $C_k = (\mathbf{C}_k(x_i, y_j))_{i,j}$ and discuss in the remark afterward how to account for the fact that our cost $C_k$ in $(\mathscr{P}_n)$ are centered.

Note that the following classical result is a direct consequence of Lemma 15

**Lemma 18.** *Suppose $Z_1, \dots, Z_m$ are independent mean zero random variables taking values in Hilbert space $\mathcal{H}$. Suppose there is some $C > 0$ such that for all $k$, $\|Z_k\| \leqslant C$. Then, for all $t > 0$, with probability at least $1 - 2\exp(-t)$,*

$$\left\| \frac{1}{m} \sum_{k=1}^{m} Z_k \right\|^2 \leqslant \frac{8C^2 t}{m}.$$

**Proposition 19.** *Let $C = \max_{x,y} \|\mathbf{C}(x, y)\|$. Let $(x_i, y_i)_{i=1}^n$ be iid drawn from $\hat{\pi}$. Then,*

$$\left\| \Phi_n^* \hat{P} - \Phi^* \hat{\pi} \right\| \leqslant \frac{\sqrt{8}Ct}{n}$$

*with probability at least $1 - 2\exp(-t^2)$.*

*Proof.* Direct consequence of Lemma 18. $\qquad\square$

**Proposition 20.** *Let $t > 0$. Let $(x_i, y_i)_{i=1}^n$ be i.i.d. drawn from $\hat{\pi}$, which has marginals $\alpha, \beta$. Let $P = \frac{1}{n^2}(p(x_i, y_j))_{i,j=1}^n$ where $\pi$ has marginals $\alpha, \beta$ and $p = \frac{d\pi}{d(\alpha \otimes \beta)}$. Then, $\mathbb{E}\left\| \Phi_n^* P - \Phi^*(p\alpha \otimes \beta) \right\|^2 = \mathcal{O}(n^{-1})$ and*

$$\left\| \Phi_n^* P - \Phi^* p\alpha \otimes \beta \right\| \lesssim \frac{1+t}{\sqrt{n}}$$

*with probability at least $1 - 2\exp(-2t^2/(64\|p\|_\infty^2))$*

*Proof.* Let $h(x, y) \triangleq p(x, y)\mathbf{C}(x, y)$, then

$$\Phi_n^* P = \frac{1}{n^2} \sum_{j=1}^{n} \left( h(x_j, y_j) + \sum_{\substack{i=1 \\ i \neq j}}^{n} h(x_i, y_j) \right)$$

and

$$\mathbb{E}[\Phi_n^* P - \Phi^* p\alpha \otimes \beta] = \frac{1}{n} A^* \left( p\hat{\pi} - p\alpha \otimes \beta \right)$$

It follows that

$$\mathbb{E}\left\| \Phi_n^* P - \Phi^* p\alpha \otimes \beta \right\|^2 = \frac{1}{n^2} \left\| \Phi^* \left( p\hat{\pi} - p\alpha \otimes \beta \right) \right\|^2 + \mathrm{Var}(\Phi_n^* P)$$

and

$$\mathrm{Var}(\Phi_n^* P) = \frac{1}{n^4} \sum_{i,j,k,\ell=1}^{n} \mathrm{Cov}(h(x_i, y_k), h(x_\ell, y_k))$$

Note that $\mathrm{Cov}(h(x_i, y_k), h(x_\ell, y_k)) = 0$ whenever $i, j, k, \ell$ are distinct and there are $n(n-1)(n-2)(n-3) = n^4 - 6n^3 + 12n^2 - 6n$ such terms, i.e. there are $6n^3 - 12n^2 + 6n$ nonzero terms in the sum. It follows that $\mathrm{Var}(\Phi_n^* P) = \mathcal{O}(n^{-1})$ if $\|\mathbf{C}\|$ and $p$ are uniformly bounded.

To conclude, we apply Lemma 15 with $f(z_1, \dots, z_n) = \left\| \Phi_n^* P - \Phi^* p\alpha \otimes \beta \right\|$ and $z_i = (x_i, y_i)$. Let $X(z_1, \dots, z_n) = \Phi_n^* P - \Phi^* p\alpha \otimes \beta$. Then, Note that for an arbitrary vector $u$,

$$\langle u, X(z_1, z_2, \dots, z_n) - X(z_1', z_2, \dots, z_n) \rangle = \sum_{i,j} \left( \sum_{k=1}^{s} u_k(h(x_i, y_j) - h(x_i', y_j')) \right)$$

$$= \sum_{i=2}^{n} \left( \sum_{k=1}^{s} u_k(h(x_i, y_1) - h(x_i', y_1)) \right) + \sum_{j=1}^{n} \left( \sum_{k=1}^{s} u_k(h(x_1, y_j) - h(x_1', y_j')) \right)$$

$$\leqslant 8n^{-1}\|u\|\sup_{x,y}\|\mathbf{C}(x,y)\|\|p\|_\infty$$

So, $f(z_1, z_2, \dots, z_n) - f(z_1', z_2, \dots, z_n) \leqslant 8n^{-1}\|p\|_\infty$ and

$$\|\Phi_n^* P - \Phi^* p\alpha \otimes \beta\| \lesssim \frac{1+t}{\sqrt{n}}$$

with probability at least $1 - 2\exp(-2t^2/(64\|p\|_\infty^2))$

$\square$

*Remark* 21 (Adjusting for the centred cost parametrization). In the above two propositions, we compare $\Phi_n^* P = \sum_{i,j} \mathbf{C}_k(x_i, y_j)P_{i,j}$ with $\Phi^*\pi = \int \mathbf{C}_k(x,y)d\pi(x,y)$ for $P$ being a discretized version of $\pi$. The delicate issue is that in $(\mathscr{P}_n)$, $C_k$ is a centralized version of $\hat{C}_k \triangleq (\mathbf{C}_k(x_i, y_j))_{i,j}$. In particular,

$$C_k = \hat{C}_k - \frac{1}{n}\sum_{i=1}^n (\hat{C}_k)_{i,j} - \frac{1}{n}\sum_{i=1}^n (\hat{C}_k)_{i,j} + \frac{1}{n^2}\sum_{j=1}^n\sum_{i=1}^n (\hat{C}_k)_{i,j}.$$

Note that if $P\mathbb{1} = \frac{1}{n}\mathbb{1}$ and $P^\top\mathbb{1} = \frac{1}{n}\mathbb{1}$, then

$$\sum_{i,j}(C_k)_{i,j}P_{i,j} = \sum_{i,j}(\hat{C}_k)_{i,j}P_{i,j} - \frac{1}{n^2}\sum_{j=1}^n\sum_{k=1}^n (\hat{C}_k)_{k,j}$$

This last term on the RHS is negligible because $\Phi^*(\alpha \otimes \beta) = 0$ by assumption of $\mathbf{C}_k$ being centred: Note that

$$\frac{1}{n^2}\sum_{j=1}^n\sum_{k=1}^n (\hat{C}_k)_{k,j} = \hat{\Phi}_n^*(\frac{1}{n^2}\mathbb{1}_{n\times n}).$$

Applying the above proposition,

$$\left\|\frac{1}{n^2}\sum_{j=1}^n\sum_{k=1}^n (\hat{C}_k)_{k,j} - \Phi^*(\alpha \otimes \beta)\right\| \lesssim \frac{m + \log(n)}{\sqrt{n}}$$

with probability at least $1 - \exp(-m)$. So, up to constants, the above two propositions can be applied even for our centralized cost $C$.

### D.2.2 INVERTIBILITY BOUND

Recall our assumption that $M \triangleq \mathbb{E}_{(x,y)\sim\alpha\otimes\beta}[\mathbf{C}(x,y)\mathbf{C}(x,y)^\top]$ is invertible. In Proposition 23, we bound the deviation of $\Phi_n^*\Phi_n$ from $M$ in the spectral norm, and hence establish that it is invertible with high probability.

We will make use of the following matrix Bernstein inequality.

**Theorem 22** (Matrix Bernstein). *Tropp et al. (2015) Let $Z_1, \dots, Z_n \in \mathbb{R}^{d\times d}$ be independent symmetric mean-zero random matrices such that $\|Z_i\| \leqslant L$ for all $i \in [n]$. Then, for all $t \geqslant 0$,*

$$\mathbb{E}\left\|\sum_i Z_i\right\| \leqslant \sqrt{2\sigma\log(2d)} + \frac{1}{3}L\log(2d)$$

*where $\sigma^2 = \left\|\sum_{i=1}^n \mathbb{E}[Z_i^2]\right\|$.*

**Proposition 23.** *Assume that $\log(2s) + 1 \leqslant n$. Let $t > 0$. Then,*

$$\left\|\frac{1}{n^2}\Phi_n^*\Phi_n - M\right\| \lesssim \sqrt{\frac{\log(2s)}{n-1}} + \frac{t}{\sqrt{n}}$$

*with probability at least $1 - \mathcal{O}(\exp(-t^2))$.*

*Proof.* Recall that $\Phi_n A = \sum_k A_k C_k$, where $C_k$ is the centred version of the matrix $\hat{C}_k = (\mathbf{C}_k(x_i, y_j))_{i,j}$. That is,

$$(C_k)_{i,j} = (\hat{C}_k)_{i,j} - \frac{1}{n}\sum_\ell (\hat{C}_k)_{\ell,j} - \frac{1}{n}\sum_m (\hat{C}_k)_{i,m} + \frac{1}{n^2}\sum_\ell\sum_m (\hat{C}_k)_{\ell,m}. \tag{37}$$

For simplicity, we first do the proof for $(\Phi_n A)_{i,j} = \sum_k A_k \mathbf{C}_k(x_i, y_j)$, and explain at the end how to modify the proof to account for $\Phi_n$ using the centered $C_k$.

First,

$$\frac{1}{n^2}\Phi_n^*\Phi_n - M = \frac{1}{n}\sum_{i=1}^n\left(\frac{1}{n}\sum_{j=1}^n \mathbf{C}(x_i, y_j)\mathbf{C}(x_i, y_j)^\top - \int \mathbf{C}(x_i, y)\mathbf{C}(x_i, y)^\top d\beta(y)\right) \tag{38}$$

$$+ \frac{1}{n}\sum_{i=1}^n \int \mathbf{C}(x_i, y)\mathbf{C}(x_i, y)^\top d\beta(y) - \int \mathbf{C}(x, y)\mathbf{C}(x, y)^\top d\alpha(x)d\beta(y) \tag{39}$$

To bound the two terms in (39), let $Z_i \triangleq \int \mathbf{C}(x_i, y)\mathbf{C}(x_i, y)^\top d\beta(y) - \int \mathbf{C}(x, y)\mathbf{C}(x, y)^\top d\alpha(x)d\beta(y)$ and observe that these are i.i.d. matrices with zero mean. By matrix Bernstein with the bounds $\|Z_i\| \le 2$ and $\|Z_i^2\| \le 4$,

$$\mathbb{E}\left\|\frac{1}{n}\sum_{i=1}^n Z_i\right\| \le \frac{\sqrt{8\log(2s)}}{\sqrt{n}} + \frac{2\log(2s)}{3n} \le 4\sqrt{\frac{\log(2s)}{n}}$$

assuming that $\log(2s) \le n$.

For the two terms in (38),

$$\mathbb{E}\left\|\frac{1}{n}\sum_{i=1}^n\left(\frac{1}{n}\sum_{j=1}^n \mathbf{C}(x_i, y_j)\mathbf{C}(x_i, y_j)^\top - \int \mathbf{C}(x_i, y)\mathbf{C}(x_i, y)^\top d\beta(y)\right)\right\| \tag{40}$$

$$\le \mathbb{E}\left\|\frac{1}{n^2}\sum_{i=1}^n \mathbf{C}(x_i, y_i)\mathbf{C}(x_i, y_i)^\top\right\| + \mathbb{E}\left\|\frac{1}{n}\sum_{i=1}^n\left(\frac{1}{n}\sum_{\substack{j=1\\j\neq i}}^n \mathbf{C}(x_i, y_j)\mathbf{C}(x_i, y_j)^\top - \int \mathbf{C}(x_i, y)\mathbf{C}(x_i, y)^\top d\beta(y)\right)\right\| \tag{41}$$

$$\le \frac{2}{n} + \frac{n-1}{n^2}\sum_{i=1}^n \mathbb{E}\left\|\frac{1}{n-1}\sum_{\substack{j=1\\j\neq i}}^n\left(\mathbf{C}(x_i, y_j)\mathbf{C}(x_i, y_j)^\top - \int \mathbf{C}(x_i, y)\mathbf{C}(x_i, y)^\top d\beta(y)\right)\right\| \tag{42}$$

For each $i = 1, \ldots n$, let $Y_j = \mathbf{C}(x_i, y_j)\mathbf{C}(x_i, y_j)^\top - \int \mathbf{C}(x_i, y)\mathbf{C}(x_i, y)^\top d\beta(y)$ and observe that conditional on $x_i$, $\{Y_j\}_{j\neq i}$ are iid matrices with zero mean. The matrix Bernstein inequality applied to $\frac{1}{n-1}\sum_{j\in[n]\setminus\{i\}} Y_j$ implies that

$$\mathbb{E}\left\|\frac{1}{n-1}\sum_{j\in[n]\setminus\{i\}} Y_j\right\| \le \frac{\sqrt{8\log(2s)}}{\sqrt{n-1}} + \frac{2\log(2s)}{3n-3} \le 4\sqrt{\frac{\log(s)}{n}}$$

assuming that $\log(2s) \le n-1$.

Finally, we apply Lemma 15 to $f(z_1, \ldots, z_n) = \left\|\frac{1}{n^2}\Phi_n^*\Phi_n - M\right\|$. Let $\Phi_n' u = \sum_k u_k \mathbf{C}(x_i', y_i')$ with $x_i' = x_i$ and $y_i' = y_i$ for $i \ge 2$. For each vector $u$ of unit norm,

$$\frac{1}{n^2}\langle(\Phi_n^*\Phi_n - (\Phi_n')^*\Phi_n')u, u\rangle = \frac{1}{n^2}\sum_{i=1}^n\sum_{j=1}^n \left|\mathbf{C}(x_i, y_j)^\top u\right|^2 - \left|\mathbf{C}(x_i', y_j')^\top u\right|^2 \tag{43}$$

$$= \frac{1}{n^2}\sum_{i=1}^n \left|\mathbf{C}(x_i, y_1)^\top u\right|^2 - \left|\mathbf{C}(x_i', y_1')^\top u\right|^2 + \frac{1}{n^2}\sum_{j=2}^n \left|\mathbf{C}(x_1, y_j)^\top u\right|^2 - \left|\mathbf{C}(x_1', y_j')^\top u\right|^2 \le 4n^{-1}. \tag{44}$$

So,

$$\left\|\frac{1}{n^2}\Phi_n^*\Phi_n - M\right\| \le 8\sqrt{\frac{\log(2s)}{n-1}} + \frac{t}{\sqrt{n}}$$

with probability at least $1 - \exp(-2t^2/16)$.

To conclude this proof, we now discuss how to modify the above proof in the case of the centered cost $C_k$, that is $\Phi_n A = \sum_k A_k C_k$ where $C_k$ is as defined in (37). Note that in this case,

$$\frac{1}{n^2}(\Phi_n^* \Phi_n)_{k,\ell} = \frac{1}{n^2} \sum_{i,j=1}^{n} \mathbf{C}_k(x_i, y_j)\mathbf{C}_\ell(x_i, y_j) + \frac{1}{n^4}\left(\sum_{p,q=1}^{n} \mathbf{C}_\ell(x_p, y_q)\right)\left(\sum_{i,j=1}^{n} \mathbf{C}_k(x_i, y_j)\right) \tag{45}$$

$$- \frac{1}{n^3}\sum_{j=1}^{n}\left(\sum_{p=1}^{n}\mathbf{C}_\ell(x_p, y_j)\right)\left(\sum_{i=1}^{n}\mathbf{C}_k(x_i, y_j)\right) - \frac{1}{n^3}\sum_{i=1}^{n}\left(\sum_{q=1}^{n}\mathbf{C}_\ell(x_i, y_q)\right)\left(\sum_{j=1}^{n}\mathbf{C}_k(x_i, y_j)\right) \tag{46}$$

We already know from the previous arguments that

$$\mathbb{E}\left\|\frac{1}{n^2}\sum_{i,j=1}^{n}\mathbf{C}_k(x_i, y_j)\mathbf{C}_\ell(x_i, y_j) - M\right\| = \mathcal{O}(\sqrt{\log(s)/n}).$$

For the last term in (45), let $\Lambda = \left\{(i, j, p, q) \,;\, i \in [n], j \in [n] \setminus \{i\}, \, p \in [n] \setminus \{i, j\}, q \in [n] \setminus \{i, j, p\}\right\}$. Note that $\Lambda$ has $n(n-1)(n-2)(n-3)$ terms, and $\Lambda^c = [n] \times [n] \times [n] \times [n]$ has $\mathcal{O}(n^3)$ terms. Therefore, we can write

$$\mathbb{E}\left\|\frac{1}{n^4}\left(\sum_{p,q=1}^{n}\mathbf{C}(x_p, y_q)\right)\left(\sum_{i,j=1}^{n}\mathbf{C}(x_i, y_j)^\top\right)\right\|$$

$$\leq \frac{1}{n^4}\mathbb{E}\left\|\sum_{(i,j,p,q)\in\Lambda}\mathbf{C}(x_p, y_q)\mathbf{C}(x_i, y_j)^\top\right\| + \mathbb{E}\left\|\frac{1}{n^4}\sum_{(i,j,p,q)\in\Lambda^c}\mathbf{C}(x_p, y_q)\mathbf{C}(x_i, y_j)^\top\right\|$$

$$\leq \frac{1}{n^4}\mathbb{E}\left\|\sum_{(i,j,p,q)\in\Lambda}\mathbf{C}(x_p, y_q)\mathbf{C}(x_i, y_j)^\top\right\| + \mathcal{O}(n^{-1})$$

where the second inequality is because there are $\mathcal{O}(n^3)$ terms in $\Lambda^c$ and we used the bound that $\|\mathbf{C}(x, y)\| = 1$. Moreover,

$$\frac{1}{n^4}\mathbb{E}\left\|\sum_{(i,j,p,q)\in\Lambda}\mathbf{C}(x_p, y_q)\mathbf{C}(x_i, y_j)^\top\right\| \leq \frac{n-3}{n^4}\sum_{i}\sum_{j\notin\{i\}}\sum_{p\notin\{i,j\}}\mathbb{E}\left\|\frac{1}{n-3}\sum_{q\notin\{i,j,p\}}\mathbf{C}(x_p, y_q)\mathbf{C}(x_i, y_j)^\top\right\|.$$

Note that conditional on $i, j, p$,

$$\mathbb{E}[\frac{1}{n-3}\sum_{q\notin\{i,j,p\}}\mathbf{C}(x_p, y_q)\mathbf{C}(x_i, y_j)^\top] = \int \mathbf{C}(x_p, y)d\beta(y)\mathbf{C}(x_i, y_j)^\top = 0$$

by assumption that $\mathbf{C}(x, y)d\beta(y) = 0$. So, we can apply Matrix Bernstein as before to show that

$$\mathbb{E}[\frac{1}{n^4}\mathbb{E}\left\|\sum_{(i,j,p,q)\in\Lambda}\mathbf{C}(x_p, y_q)\mathbf{C}(x_i, y_j)^\top\right\|] \lesssim \sqrt{\log(s)\, n^{-1}}.$$

A similar argument can be applied to handle the two terms in (46), and so,

$$\mathbb{E}\left\|\frac{1}{n^2}\Phi_n^* \Phi_n - M\right\| \lesssim \sqrt{\log(s)/n}.$$

The high probability bound can now be derived using Lemma 15 as before. $\qquad\square$

# E  PROOFS FOR THE GAUSSIAN SETTING

**Simplified problem**  To ease the computations, we will compute the Hessian of the following function (corresponding to the special case where $\Sigma_\alpha = \mathrm{Id}$ and $\Sigma_\beta = \mathrm{Id}$):

$$\tilde{W}(A) \triangleq \sup_{\Sigma}\langle A, \Sigma\rangle + \frac{\varepsilon}{2}\log\det(\mathrm{Id} - \Sigma^\top\Sigma). \tag{47}$$

To retrieve the Hessian of the original function note that, since $\log\det(\Sigma_\beta - \Sigma^\top\Sigma_\alpha^{-1}\Sigma) = \log\det(\Sigma_\beta) + \log\det(\mathrm{Id} - \Sigma_\beta^{-\frac{1}{2}}\Sigma^\top\Sigma_\alpha^{-1}\Sigma\Sigma_\beta^{-\frac{1}{2}})$, a change of variable $\tilde{\Sigma} \triangleq \Sigma_\alpha^{-\frac{1}{2}}\Sigma\Sigma_\beta^{-\frac{1}{2}}$, shows that $W(A) = \tilde{W}(\Sigma_\alpha^{\frac{1}{2}} A\Sigma_\beta^{\frac{1}{2}})$ and, hence,

$$\nabla^2 W(A) = (\Sigma_\beta^{\frac{1}{2}} \otimes \Sigma_\alpha^{\frac{1}{2}})\nabla^2\tilde{W}(\Sigma_\alpha^{\frac{1}{2}} A\Sigma_\beta^{\frac{1}{2}})(\Sigma_\beta^{\frac{1}{2}} \otimes \Sigma_\alpha^{\frac{1}{2}}). \tag{48}$$

By the envelope theorem, $\nabla \tilde{W}(A) = \Sigma$, where $\Sigma$ is the maximizer of (47), thus reducing the computation of $\nabla^2 \tilde{W}(A)$ to differentiating the optimality condition of $\Sigma$, *i.e.,*

$$A = \varepsilon^{-1} \Sigma (\mathrm{Id} - \Sigma^\top \Sigma)^{-1}. \tag{49}$$

Recall also that such a $\Sigma$ has an explicit formula given in (5).

**Lemma 24.**

$$\nabla^2 \tilde{W}(A) = \varepsilon \left( \varepsilon^2 (\mathrm{Id} - \Sigma^\top \Sigma)^{-1} \otimes (\mathrm{Id} - \Sigma \Sigma^\top)^{-1} + (A^\top \otimes A) \mathbb{T} \right)^{-1} \tag{50}$$

*and $\Sigma$ is as in (49) and $\mathbb{T}$ is the linear map defined by $\mathbb{T} \, vec(X) = vec(X^\top)$.*

*Proof of Lemma 24.* Define $G(\Sigma, A) \triangleq \Sigma (\mathrm{Id} - \Sigma^\top \Sigma)^{-1} - \varepsilon^{-1} A$. Note that a maximizer $\Sigma$ of (47) satisfies $G(\Sigma, A) = 0$. Moreover, $\partial_\Sigma G$ is invertible at such $(\Sigma, A)$ because this is the Hessian of $\log \det(\mathrm{Id} - \Sigma^\top \Sigma)$, a strictly concave function. By the implicit function theorem, there exists a function $f : A \mapsto \Sigma$ such that $G(f(A), A) = 0$ and

$$\nabla f = (\partial_\Sigma G)^{-1} \partial_A G = \varepsilon^{-1} (\partial_\Sigma G)^{-1}.$$

It remains to compute $\partial_\Sigma G$ at $(f(A), A)$:

At an arbitrary point $(\Sigma, A)$ we have

$$\begin{aligned}
\partial_\Sigma G &= (\mathrm{Id} - \Sigma^\top \Sigma)^{-1} \otimes \mathrm{Id} + (\mathrm{Id} \otimes \Sigma) \partial_\Sigma (\mathrm{Id} - \Sigma^\top \Sigma)^{-1} \\
&= (\mathrm{Id} - \Sigma^\top \Sigma)^{-1} \otimes \mathrm{Id} + (\mathrm{Id} \otimes \Sigma) \left( (\mathrm{Id} - \Sigma^\top \Sigma)^{-1} \otimes (\mathrm{Id} - \Sigma^\top \Sigma)^{-1} \right) \left( (\Sigma^\top \otimes \mathrm{Id}) \mathbb{T} + (\mathrm{Id} \otimes \Sigma^\top) \right)
\end{aligned}$$

and at $(f(A), A)$, *i.e*, with $\Sigma (\mathrm{Id} - \Sigma^\top \Sigma)^{-1} = \varepsilon^{-1} A$, we can further simplify to

$$\partial_\Sigma G = (\mathrm{Id} - \Sigma^\top \Sigma)^{-1} \otimes (\mathrm{Id} + \Sigma (\mathrm{Id} - \Sigma^\top \Sigma)^{-1} \Sigma^\top)) + \varepsilon^{-2} (A^\top \otimes A) \mathbb{T}$$

By the Woodbury matrix formula,

$$(\mathrm{Id} + \Sigma (\mathrm{Id} - \Sigma^\top \Sigma)^{-1} \Sigma^\top) = (\mathrm{Id} - \Sigma \Sigma^\top)^{-1} = \mathrm{Id} + A \Sigma^\top.$$

So,

$$\partial_\Sigma G = (\mathrm{Id} - \Sigma^\top \Sigma)^{-1} \otimes (\mathrm{Id} - \Sigma \Sigma^\top)^{-1} + \varepsilon^{-2} (A^\top \otimes A) \mathbb{T},$$

thus concluding the proof. $\qquad \square$

We remark that from the connection between $W(A)$ and $\tilde{W}(A)$, *i.e.,* 48, we obtain Lemma 7 as a corollary.

### E.1 Limit cases

**SVD representation of the covariance**   To derive the limiting expressions, we make an observation on the singular value decomposition of $\Sigma$: Let the singular value decomposition of $A$ be $A = UDV^\top$, where $D$ is the diagonal matrix with positive entries $d_i$. Note that $\Delta = (\mathrm{Id} + \frac{\varepsilon^2}{4} (A^\top A)^\dagger)^{\frac{1}{2}} = V (\mathrm{Id} + \frac{\varepsilon^2}{4} D^{-2})^{\frac{1}{2}} V^\top$. Moreover, $\Delta$ and $A^\top A$ commute, so

$$\Sigma = A \Delta ((\Delta^2 A^\top A)^\dagger)^{\frac{1}{2}} \Delta - \frac{\varepsilon}{2} A^{T,\dagger} = U \left( \left( \mathrm{Id} + \frac{\varepsilon^2}{4} D^{\dagger,2} \right)^{\frac{1}{2}} - \frac{\varepsilon}{2} D^\dagger \right) V^\top = U \tilde{D} V^\top, \tag{51}$$

where $\tilde{D}$ is the diagonal matrix with diagonal entries

$$\tilde{d}_i = \sqrt{\left( 1 + \frac{\varepsilon^2}{4 d_i^2} \right)} - \frac{\varepsilon}{2} \frac{1}{d_i}. \tag{52}$$

### E.1.1 Link with Lasso: Limit as $\varepsilon \to \infty$

Note that

$$\tilde{d}_i = \frac{\varepsilon}{2d_i}\sqrt{1 + \frac{4d_i^2}{\varepsilon^2}} - \frac{\varepsilon}{2d_i} = \frac{d_i}{\varepsilon} - \frac{d_i^3}{\varepsilon^3} + \mathcal{O}(\varepsilon^{-5}) \to 0, \qquad \varepsilon \to \infty. \tag{53}$$

It follows that $\lim_{\varepsilon \to \infty} \Sigma = 0$ and hence, $\varepsilon \nabla^2 \tilde{W}(A) \to \mathrm{Id}$. So, the certificate converges to

$$(\Sigma_\beta \otimes \Sigma_\alpha)_{(:,I)}\big((\Sigma_\beta \otimes \Sigma_\alpha)_{(I,I)}\big)^{-1}\mathrm{sign}(\widehat{A})_I. \tag{54}$$

*Proof of Proposition 8.* The iOT problem approaches a Lasso problem as $\varepsilon \to \infty$. Recall that in the Gaussian setting, the iOT problem is of the form

$$\underset{A}{\arg\min}\,\mathcal{F}(A) = \lambda\,\|A\|_1 + \langle A, \Sigma_{\varepsilon,A} - \Sigma_{\varepsilon,\widehat{A}}\rangle + \frac{\varepsilon}{2}\log\det\big(\Sigma_\beta - \Sigma_{\varepsilon,A}^\top\Sigma_\alpha^{-1}\Sigma_{\varepsilon,A}\big) \tag{55}$$

where $\Sigma_{\varepsilon,A}$ satisfies $\Sigma_\beta - \Sigma_{\varepsilon,A}^\top\Sigma_\alpha^{-1}\Sigma_{\varepsilon,A} = \varepsilon\Sigma_\alpha^{-1}\Sigma_{\varepsilon,A}A^{-1}$. So, we can write

$$\underset{A}{\arg\min}\,\mathcal{F}(A) \triangleq \lambda\,\|A\|_1 + \langle A, \Sigma_{\varepsilon,A} - \Sigma_{\varepsilon,\widehat{A}}\rangle + \frac{\varepsilon}{2}\log\det\big(\varepsilon\Sigma_\alpha^{-1}\Sigma_{\varepsilon,A}A^{-1}\big) \tag{56}$$

Let

$$X \triangleq \Sigma_\alpha^{\frac{1}{2}}A\Sigma_\beta^{\frac{1}{2}} \quad\text{and}\quad \tilde{\Sigma}_{\varepsilon,A} = \Sigma_\alpha^{-\frac{1}{2}}\Sigma_{\varepsilon,A}\Sigma_\beta^{-\frac{1}{2}}.$$

From (52), if $X$ has SVD decomposition $W = U\mathrm{diag}(d_i)V^\top$, then

$$\tilde{\Sigma}_{\varepsilon,A} = U\tilde{D}V^\top, \quad\text{where}\quad \tilde{D} = \mathrm{diag}(\tilde{d}_i)$$

and

$$\tilde{d}_i = \frac{d_i}{\varepsilon} - \frac{d_i^3}{\varepsilon^3} + \mathcal{O}(\varepsilon^{-5}).$$

So,

$$\begin{aligned}
\log\det(\varepsilon\Sigma_\alpha^{-1}\Sigma_{\varepsilon,A}A^{-1}) &= \log\det\left(\varepsilon\Sigma_\alpha^{-\frac{1}{2}}\tilde{\Sigma}_{\varepsilon,A}X^{-1}\Sigma_\alpha^{\frac{1}{2}}\right)\\
&= \log\det\big(U\mathrm{diag}\big(1 - d_i^2/\varepsilon^2 + \mathcal{O}(\varepsilon^{-4})\big)U^\top\big)\\
&= \log\det\big(\mathrm{diag}\big(1 - d_i^2/\varepsilon^2 + \mathcal{O}(\varepsilon^{-4})\big)\big) = -\frac{1}{\varepsilon^2}\|X\|_F^2 + \mathcal{O}(\varepsilon^{-4}).
\end{aligned}$$

Also,

$$\varepsilon\langle\Sigma_{\varepsilon,A} - \Sigma_{\varepsilon,\widehat{A}}, A\rangle = \varepsilon\langle\tilde{\Sigma}_{\varepsilon,A} - \tilde{\Sigma}_{\varepsilon,\widehat{A}}, X\rangle = \langle X - X_0, X\rangle + \mathcal{O}(\varepsilon^{-2}).$$

So, assuming that $\lambda = \lambda_0/\varepsilon$,

$$\varepsilon\mathcal{F}(A) = \lambda_0\,\|A\|_1 + \langle X - X_0, X\rangle - \|X\|_F^2 + \mathcal{O}(\varepsilon^{-2}) \tag{57}$$

$$= \lambda_0\,\|A\|_1 + \left\|(\Sigma_\beta^{\frac{1}{2}} \otimes \Sigma_\alpha^{\frac{1}{2}})(A - \widehat{A})\right\|_F^2 - \frac{1}{2}\left\|(\Sigma_\beta^{\frac{1}{2}} \otimes \Sigma_\alpha^{\frac{1}{2}})\widehat{A}\right\|^2 + \mathcal{O}(\varepsilon^{-2}) \tag{58}$$

The final statement on the convergence of minimizers follows by Gamma-convergence. $\qquad\square$

### E.1.2 Link with Graphical Lasso

In the special case where the covariances are the identity ($\Sigma_\alpha = \mathrm{Id}$ and $\Sigma_\beta = \mathrm{Id}$) and $A$ is symmetric positive definite we have that $\Sigma$ is also positive definite and Galichon's formula (5) holds (since $A$ is invertible) and hence the Hessian reduces to

$$\left(\frac{1}{\varepsilon}\nabla^2\tilde{W}(A)\right)^{-1} = (A \otimes A)\big(\Sigma^{-1} \otimes \Sigma^{-1} + \mathbb{T}\big). \tag{59}$$

Moreover, if $A$ admits an eigenvalue decomposition $A = UDU^\top$, then $\Sigma$ admits an eigenvalue decomposition $\Sigma = U\tilde{D}U^\top$ with entries of $\tilde{D}$ given by (52). Note that it follows that $\lim_{\varepsilon \to 0}\Sigma = \mathrm{Id}$ and, hence, $\lim_{\varepsilon \to 0}\big(\Sigma^{-1} \otimes \Sigma^{-1} + \mathbb{T}\big) = \mathrm{Id} + \mathbb{T}$, a singular matrix with the kernel being the set of

asymmetric matrices. So, the limit does not necessarily exist as $\varepsilon \to 0$. However, in the special case where $A$ is *symmetric positive definite*, one can show that the certificates remain well defined as $\varepsilon \to 0$: Let $S_+$ be the set of matrices $\psi$ such that $\psi$ is symmetric and $S_-$ be the set of matrices $\psi$ such that $\psi$ is anti-symmetric. Then, $(\nabla^2 \tilde{W}(A))^{-1}(S_+) \subset S_+$ and $(\nabla^2 \tilde{W}(A))^{-1}(S_-) \subset S_-$. Moreover,

$$\left(\frac{1}{\varepsilon}\nabla^2 \tilde{W}(A)\right)^{-1} \restriction_{S_+} = (A \otimes A)\left(\Sigma^{-1} \otimes \Sigma^{-1} + \mathrm{Id}\right),$$

$$\left(\frac{1}{\varepsilon}\nabla^2 \tilde{W}(A)\right)^{-1} \restriction_{S_-} = (A \otimes A)\left(\Sigma^{-1} \otimes \Sigma^{-1} - \mathrm{Id}\right).$$

Since the symmetry of $A$ implies the symmetry of $\mathrm{sign}(A)$, we can replace the Hessian given in (59) by $(A \otimes A)(\Sigma^{-1} \otimes \Sigma^{-1} + \mathrm{Id})$ and, hence, since $\lim_{\varepsilon \to 0} \Sigma = \mathrm{Id}$, the limit as $\varepsilon \to 0$ is

$$\lim_{\varepsilon \to 0} z_\varepsilon = (A^{-1} \otimes A^{-1})_{(:,I)}\left((A^{-1} \otimes A^{-1})_{(I,I)}\right)^{-1} \mathrm{sign}(A)_I. \tag{60}$$

This coincides precisely with the certificate of the *graphical lasso*:

$$\operatorname*{arg\,min}_{\Theta \geq 0} \langle S, \Theta \rangle - \log \det(\Theta) + \lambda \|\Theta\|_1.$$

*Proof of Proposition 9.* The iOT problem with identity covariances restricted to the set of positive semi-definite matrices has the form

$$\operatorname*{arg\,min}_{A \geq 0} \mathscr{F}_{\varepsilon,\lambda}(A), \quad \text{where} \quad \mathscr{F}_{\varepsilon,\lambda}(A) \triangleq \lambda \|A\|_1 + \langle A, \Sigma_{\varepsilon,A} - \hat{\Sigma} \rangle + \frac{\varepsilon}{2} \log \det(\mathrm{Id} - \Sigma_{\varepsilon,A}^\top \Sigma_{\varepsilon,A}), \tag{61}$$

where $I - \Sigma_{\varepsilon,A}^\top \Sigma_{\varepsilon,A} = \varepsilon \Sigma_{\varepsilon,A} A^{-1}$. From the singular value decomposition of $\Sigma_{\varepsilon,A}$, *i.e.*, (51), we see that if $A$ is symmetric positive definite, then so is $\Sigma_{\varepsilon,A}$. Plugging the optimality condition, *i.e.*, $I - \Sigma_{\varepsilon,A}^\top \Sigma_{\varepsilon,A} = \varepsilon \Sigma_{\varepsilon,A} A^{-1}$, into (61), we obtain

$$\operatorname*{arg\,min}_{A \geq 0} \lambda \|A\|_1 + \langle A, \Sigma_{\varepsilon,A} - \hat{\Sigma} \rangle + \frac{\varepsilon}{2} \log \det(\varepsilon A^{-1} \Sigma_{\varepsilon,A})$$

$$= \operatorname*{arg\,min}_{A \geq 0} \lambda \|A\|_1 - \frac{\varepsilon}{2} \log \det(A/\varepsilon) + \varepsilon \langle A, (\Sigma_{\varepsilon,A} - \hat{\Sigma})/\varepsilon \rangle + \frac{\varepsilon}{2} \log \det(\Sigma_{\varepsilon,A})$$

Let $\lambda = \varepsilon \lambda_0$ for some $\lambda_0 > 0$. Then, removing the constant $\frac{\varepsilon}{2} \log \det(\varepsilon \mathrm{Id})$ term and factoring out $\varepsilon$, the problem is equivalent to

$$\operatorname*{arg\,min}_{A \geq 0} \lambda_0 \|A\|_1 - \frac{1}{2} \log \det(A) + \langle A, \varepsilon^{-1}(\Sigma_{\varepsilon,A} - \hat{\Sigma}) \rangle + \frac{1}{2} \log \det(\Sigma_{\varepsilon,A})$$

Assume that $\hat{\Sigma} = \Sigma_{\varepsilon,\hat{A}}$. From the expression for the singular values of $\Sigma_{\varepsilon,A}$ in (52), note that

$$\Sigma_{\varepsilon,A} = \mathrm{Id} - \frac{\varepsilon}{2} A^{-1} + \mathcal{O}(\varepsilon^2).$$

So, $\lim_{\varepsilon \to 0}(\Sigma_{\varepsilon,A} - \Sigma_{\varepsilon,\hat{A}})/\varepsilon = -\frac{1}{2}\left(A^{-1} - \hat{A}^{-1}\right)$. The objective converges pointwise to

$$\lambda_0 \|A\|_1 - \frac{1}{2} \log \det(A) + \frac{1}{2}\langle A, \hat{A}^{-1} - A^{-1} \rangle,$$

and the statement is then a direct consequence of Gamma-convergence. $\square$

*Remark* 25. Note that from (52), we have $\Sigma = \mathrm{Id} - \frac{\varepsilon}{2} A^{-1} + \mathcal{O}(\varepsilon^2)$. So, in this case, the covariance of $\hat{\pi}$ is

$$\begin{pmatrix} \mathrm{Id} & \mathrm{Id} - \frac{\varepsilon}{2} A^{-1} + \mathcal{O}(\varepsilon^2) \\ \mathrm{Id} - \frac{\varepsilon}{2} A^{-1} + \mathcal{O}(\varepsilon^2) & \mathrm{Id} \end{pmatrix}.$$

For $(X, Y) \sim \pi$, The Schur complement of this is the covariance of $X$ conditional on $Y$, which is $\varepsilon A^{-1} + \mathcal{O}(\varepsilon^2)$. So, up the $\varepsilon$, one can see $A^{-1}$ as the "precision" matrix of the covariance of $X$ conditional on $Y$.

# F  LARGE SCALE $\ell^1$-IOT SOLVER

Recall that the iOT optimization problem, recast over the dual potentials for empirical measures, reads

$$\inf_{A,F,G} \frac{1}{n} \sum_i F_i + G_i + (\Phi_n A)_{i,i} + \frac{\varepsilon}{2n^2} \sum_{i,j} \exp\left(\frac{2}{\varepsilon}(F_i + G_j + (\Phi_n A)_{i,j})\right) + \lambda \|A\|_1.$$

To obtain a better-conditioned optimization problem, in line with Cuturi & Peyré (2016), we instead consider the semi-dual problem, which is derived by leveraging the closed-form expression for the optimal $G$, given $F$.

$$\inf_{A,F} \frac{1}{n} \sum_i F_i + (\Phi_n A)_{i,i} + \frac{\varepsilon}{n} \sum_i \log \frac{1}{n} \sum_j \exp\left(\frac{2}{\varepsilon}(F_i + (\Phi_n A)_{i,j})\right) + \lambda \|A\|_1.$$

Following Poon & Peyré (2021), which proposes a state-of-the-art Lasso solver, the last step is to use the following Hadamard product over-parameterization of the $\ell^1$ norm

$$\|A\|_1 = \min_{U \odot V} \frac{\|U\|_2^2}{2} + \frac{\|V\|_2^2}{2}.$$

where the Hadamard product is $U \odot V \triangleq (U_i V_i)_i$, to obtain the final optimization problem

$$\inf_{A,U,V} \frac{1}{n} \sum_i F_i + (\Phi_n(U \odot V))_{i,i} + \frac{\varepsilon}{n} \sum_i \log \frac{1}{n} \sum_j \exp\left(\frac{2}{\varepsilon}(F_i + (\Phi_n(U \odot V))_{i,j})\right) + \frac{\lambda}{2} \|U\|_2^2 + \frac{\lambda}{2} \|V\|_2^2.$$

This is a smooth optimization problem, for which we employ a quasi-Newton solver (L-BFGS). Although it is non-convex, as demonstrated in Poon & Peyré (2021), the non-convexity is benign, ensuring the solver always converges to a global minimizer, $(F^\star, U^\star, V^\star)$, of the functional. From this, one can reconstruct the cost parameter, $A^\star \triangleq U^\star \odot V^\star$.

