# OpenReview forum: "Sparsistency for inverse optimal transport"
_ICLR.cc/2024/Conference — ICLR 2024 poster_

### Official Review · Reviewer_rfzP · 2023-10-31

**Soundness:** 2 fair
**Presentation:** 3 good
**Contribution:** 2 fair
**Rating:** 5
**Confidence:** 3

**Summary:**

This article addresses the problem of inverse optimal transport, which involves estimating the transport cost from an optimal (noisy) transport plan. The authors focus on the 'penalized' $\ell_1$ cost formulation, parametrized by a matrix of parameters (the cost is parametrized as linear combination of individual costs, such as in the Mahalanobis setting). The article provides two types of guarantees:

The first type of guarantee, referred to as the "finite sample case," is an estimation guarantee obtained from dual certificates when the observed transport plan is an entropic regularized plan between two empirical measures.

The second type, in the "Gaussian case," deals with the scenario where the measures involved are Gaussian distributions (a well-known case in transport that admits a closed-form solution). In this case, the article demonstrates that the cost estimation can be solved as a graphical lasso problem, especially when the entropic regularization is small.

**Strengths:**

- I find the introduction and contextualization of the article to be very well done. The related work appears comprehensive, and the problem is well-situated.

- The theoretical results in this article are interesting. For the discrete case, the authors demonstrate that, with a small $\ell_1$ regularization and a sufficiently large number of samples, the ill-posed problem of invOT can indeed recover the costs.

- The connections between graphical lasso and invOT are also intriguing and interesting.

**Weaknesses:**

- I find that the article doesn't put in enough effort to explain the practical implications of the various theoretical results, which remain somewhat abstract. It's quite challenging to understand how these results can be practically applied.

Firstly, the concept of a pre-certificate condition is rather very abstract (unlike the case of Lasso that we can somehow interpret). It would be interesting to provide the reader with a bit more guidance to offer a small interpretation of this quantity.

Another example is Proposition 8. It's difficult for me to extract something from this result, aside from the qualitative interpretation that "for small regularization, under somewhat abstract assumptions of certificates, and with a sufficiently large number of samples, solving the invOT problem yields a good solution." To make these results more applicable in practice, it would be helpful to either provide experiments or guarantees that demonstrate how to understand these pre-certificate assumptions or ensure that the level of regularization is appropriate. I realize that these may be challenging questions, but I find that not much intuition is given, and the practical use of these results does not appear straightforward.

- My main criticism concerns the experiments section. I find the experiments too limited and somewhat confusing, making it difficult to obtain meaningful information.

First, only the Gaussian with an identity covariance is considered. The idea is to study the impact of entropic regularization on invOT estimation. The presentation is somewhat unclear, and I struggle to understand from Figure 1 how it quantifies the influence of $\epsilon$ on the estimation. There is no legend for the y-axis, and while it's understood to be the certificate value between two nodes, it's unclear how it serves as a good performance measure for the invOT problem.

Figure 2 is equally unenlightening. It aims to determine the influence of the number of samples on the estimation in the very simple case of a circular graph and Gaussian measures. However, what is the x-axis on these three figures? Is it the number of iterations? The geodesic distance? If it's the geodesic distance, it's not clear because the y-axis is a global performance measure over C, while the x-axis appears to be a local measure, so it's unclear how one evolves with the other.

The conclusion appears to be that the estimation is "good" when the entropic regularization is sufficiently large, and the number of samples is also large. More importantly, there seems to be a significant gap: when $\epsilon \leq 10$, the estimation is consistently poor. Is this a result expected by the theory regarding $\epsilon$?

**Questions:**

I'm curious to know whether the Gaussian results really require entropic regularization. Without regularization we also have a closed form: can't we deduce good invOT estimation guarantees in the non-regularized case?

---

> ### Author Response · Authors · 2023-11-14
>
> Thank you for your comments.
>
> > I find that the article doesn't put in enough effort to explain the practical implications of the various theoretical results [...] to provide the reader with a bit more guidance to offer a small interpretation of this quantity.
>
>
> We have added an ``intuitions'' section to explain the relation between the certificate and optimality conditions. There is also now an "interpretation" after Propositions 8 and 9.
>
> > Another example is Proposition 8. It's difficult for me to extract something from this result,  [...] I realize that these may be challenging questions, but I find that not much intuition is given, and the practical use of these results does not appear straightforward.
>
> Proposition 8 is the main technical result required for proving Theorem 3 which tells us that sparsistency is possible under the abstract assumption on the certificate. Theoretically proving whether the certificate is nondegenerate is challenging and in this paper, we analyse this condition only in the Gaussian setting (Prop 8 and 9 in the updated version): Some practical insights is that in the small epsilon regime, recoverability is equivalent to recoverability in the graphical lasso setting while for large epsilon, recoverability is approximated by the lasso setting (e.g. diagonal covariance matrix in alpha and beta will always lead to this condition being satisfied). We added comments on ``interpretation'' after Prop 8 and 9. In our opinion, the result here is inline with general intuition that small epsilon leads to more dependent couplings (and hence graphical lasso in the limit) while large epsilon leads to independent couplings, so the interaction matrix recovers has a different interpretation in each setting.
>
> > My main criticism concerns the experiments section. I find the experiments too limited and somewhat confusing, [...]  There is no legend for the y-axis, and while it's understood to be the certificate value between two nodes, it's unclear how it serves as a good performance measure for the invOT problem.
>
> The purpose of Figure 1 was to numerically illustrate the behaviour of certificates in the Gaussian setting. One of the messages of our analysis is that in the large epsilon regime, the OT coupling becomes increasingly independent and iOT approximates a Lasso problem; in the small epsilon regime, the OT coupling becomes more dependent and iOT approximates graphical Lasso which is ``harder'' for sparsistency. The identity covariance is chosen because in this case, as $\epsilon$ increases, our theory tells us that the certificate becomes nondegenerate so sparsistency will be guaranteed. We added also another figure to illustrate the setting where one attempts to recover a nonsymmetric cost function: the connection to graphical lasso is only in the case of symmetric matrices A and when A is not symmetric, the certificate in fact is not guaranteed to have a limit as epsilon goes to 0; this corroborates with numerical observations made in previous works where it has been noted that the recovery of non-symmetric costs is harder (page 23 https://arxiv.org/pdf/2002.09650.pdf and section 4.4 of https://arxiv.org/pdf/1905.03950.pdf).
>
>
> > Figure 2 is equally unenlightening. It aims to [...] what is the x-axis on these three figures? Is it the number of iterations? The geodesic distance? If it's the geodesic distance, [...] it's unclear how one evolves with the other.
>
> We have added labels to the axis. The x-axis is $\lambda$. For each regularisation parameter $\lambda$, we solve iOT and record the number of wrongly estimated positions.
>
>
> > The conclusion appears to be that the estimation is ``good'' when the entropic regularization  is sufficiently large [...] this a result expected by the theory regarding $\epsilon$?
>
> When $\epsilon$ is small, the certificate is degenerate and hence support recovery is unstable. We added a remark on this: `` in particular,
> one can observe from (7) that for large $\epsilon$, the certificate is trivially non-degenerate whenever
> $\Sigma_\alpha$, $\Sigma_\beta$ are diagonal."
>
>
> > I'm curious to know whether the Gaussian results really require entropic regularization. Without regularization we also have a closed form: can't we deduce good invOT estimation guarantees in the non-regularized case?
>
> It is true in the $\epsilon=0$ case that the OT coupling has a closed form solution, however, it is unclear how to directly estimate the matrix A from the sample covariance matrix.
> For instance, the $\ell^1$--iOT method collapses in this case: the optimal transport function $W(A)$ becomes 1-homogeneous in $A$ when $\epsilon=0$, and so,  the loss function we have $L(A,\hat \pi)$ is also 1-homogeneous. As a result, if one considers $L(A,\hat \pi) + \lambda\|{A}\|_1$, zero is trivially a solution. So, the $\epsilon=0$ case is challenging because it is unclear if one can apply convex regularization to handle the ill-posedness of the problem.

---

### Official Review · Reviewer_2C9j · 2023-11-01

**Soundness:** 3 good
**Presentation:** 3 good
**Contribution:** 3 good
**Rating:** 8
**Confidence:** 3

**Summary:**

This paper deals with the problem of inverse optimal transport on compact (but not necessarily finite) state space to find the cost function from a given empirical probability coupling. It establishes a connection to lasso and produces the sample complexity of solving the corresponding primal-dual method.

**Strengths:**

Finding the cost function from the empirical process induced by the optimal transport algorithm is an important open question. In that respect, this paper addresses an important open question. The connection of the non-degenerate precertificate assumption with irrepresentability assumption in lasso is an interesting insight. Finally, I found the connections to the SVD of the covariance matrix illuminating. The gaussian example also serves to demonstrate the theory through the lenses of an example.

**Weaknesses:**

Although the sample complexity bound is good, there was no discussion about the tightness of the said bound. I wonder if the $1/\sqrt{n}$ is also the best lower bound. Observing that the empirical density acts like a ``plug-in" for the unknown true coupled density, can the statistical guarantees of the plug-in be translated to the guarantees for the cost function?

**Questions:**

Can the authors please elaborate more on the penalty term $\lambda$? Is $\lambda$ given for a given problem? Is it shrinking with $n$?

Also, do the authors implicitly assume that we can sample from the coupled empirical density? Or do we just have access to some samples?

---

> ### Author Response · Authors · 2023-11-14
>
> Thank you for your comments.
>
> > Although the sample complexity bound is good, there was no discussion about the tightness of the said bound. I wonder if the is also the best lower bound.
>
> Although we have no rigorous proof of this, the sample complexity of $1/\sqrt{n}$ is known to be tight for the direct estimation of eOT (https://arxiv.org/abs/1905.11882), so it should be expected that this sample complexity rate should also be tight for inverse eOT. We will comment about this in the revised version of the paper.
>
>
> > Observing that the empirical density acts like a ``plug-in" for the unknown true coupled density, can the statistical guarantees of the plug-in be translated to the guarantees for the cost function?
> > Also, do the authors implicitly assume that we can sample from the coupled empirical density? Or do we just have access to some samples?
>
> We assume that we have $n$ iid samples from some probability coupling $\hat\pi$, which is an eOT coupling between $\alpha$ and $\beta$ for some $\epsilon>0$ with underlying cost $c_{\hat A}$. The question we are addressing is whether $\hat A$ can be stably recovered from $n$ samples.
>
> To clarify, we do not observe $\hat \pi = \mathrm{Sink}(c_{\hat A}, \epsilon)$ but only $n$ iid samples $(x_i,y_i)\sim \hat \pi$ and our main result is concerned with the approximation of $\hat A$ (and hence $c_{\hat A}$) by solving the iOT problem with the plug in $\hat \pi_n = \frac{1}{n} \sum_{i=1}^n \delta_{(x_i,y_i)}$. We have modified Section 2.2 to clarify this and moved some of the technical details on how to set up the finite dimensional problem to the appendix.
>
>
> > Questions:
> Can the authors please elaborate more on the penalty term $\lambda$? Is given for a given problem? Is it shrinking with ?
>
>
> $\lambda$ is a regularization parameter in our inverse problem which should be balanced with the "noise" (in this case, number of samples) for accurate reconstructions. The relation between lambda and $n$ is given in Theorem 5 where we see that $e^{C/\lambda}/\lambda \leq  \sqrt{n}$. So, the smaller lambda is, the larger $n$ needs to be.

---

### Official Review · Reviewer_akqu · 2023-11-03

**Soundness:** 4 excellent
**Presentation:** 4 excellent
**Contribution:** 4 excellent
**Rating:** 8
**Confidence:** 2

**Summary:**

The paper studies the regularized inverse entropic optimal transport problem. Inverse optimal transport (iOT) is the problem of recovering the ground cost given samples from the (potentially, noisy) joint distribution. Recent works have proposed solvers to solve the primal and dual formulations of the iOT and regularized iOT problem. This paper studies the recovery guarantees for the L1-regularized iOT problem. They further explore a special case where the densities are Gaussian, and show that in some cases, iOT results in the graphical LASSO problem.

**Strengths:**

- Inverse optimal transport is an interesting problem and an interesting take on the metric learning problem, this work takes a significant step forward in establishing a theoretical grounding for the regularized iOT problem.
- I really enjoyed reading the paper, the writing is very clear, the background, method, and the results are well presented.
- Showing that graphical LASSO as a special case of inverse OT is very interesting and makes sense.

**Weaknesses:**

- Nothing that I can think of.

**Questions:**

- How practical is inverse OT? I understand that the current work considers linear cost functions. Can one, albeit without strong theoretical guarantees, learn a general (perhaps, regular) cost function from the samples drawn from the coupling?

---

> ### Author Response · Authors · 2023-11-14
>
> Thank you for your comments.
>
> >  How practical is inverse OT? I understand that the current work considers linear cost functions. Can one, albeit without strong theoretical guarantees, learn a general (perhaps, regular) cost function from the samples drawn from the coupling?
>
> iOT can indeed be applied to recover other types of cost functions. There are works in this direction such as https://arxiv.org/abs/2002.09650 which considers cost functions and Kantorovich potentials given by neural networks.  We have cited several of these works in the introduction, but to clarify, we also added a footnote on page 4. Note however that, in this setting it is far from trivial to give theoretical guarantees, our main goal with this work.

---

### Official Review · Reviewer_gfhu · 2023-11-03

**Soundness:** 3 good
**Presentation:** 1 poor
**Contribution:** 3 good
**Rating:** 6
**Confidence:** 2

**Summary:**

The paper is about inverse optimal transport (iOT), that is the task of inferring a ground cost from the sampling of an (entropy-regularized) optimal transport plan.
For linear parametrizations of the cose, iOT is a convex inverse problem (Dupuy et al 2019), that the authors propose to regularize with the $\ell_1$ penalty.

The paper first derives an irrepresentability condition (IC) for the iOT problem,
based on non degeneracy of a "precertificate" (Def 1).
First, the authors show that the solution of the regularized "infinite samples" problem shares the same sign as the true parameters under IC, for low enough regularization value (Thm 3).
Then the more interesting "Sparsistency" results are then derived: in the finite sample case, the correct support is still recovered (under IC) for small regularization values and sufficiently many samples (Theorem 7).

Additional details in the case of Gaussian distributions are provided, with interpretation as Lasso and Graphical Lasso respectively for vanishing or exploding entropic regularization strength.
Experiments on limited size graphs (80 nodes) conclude this theoretical paper.

**Strengths:**

Inverse optimal transport has recently attracted attention in the community due to its potential impact in ML.
Proposing better alternatives to solve this nonlinear inverse problem is thus of interest.
The paper provides results on both "full distribution" and finite sample problems.
I did not spot any mathematical error, but could not check all of the paper.

**Weaknesses:**

The authors could really afford to improve the pedagogy of the paper, which is quite heavy in terms of notation. Theory is quite involved, require background references to other works such as Carlier or Galichon.
The experiment description is a big block which, in my opinion, does not bring as many insights as it could.

**Questions:**

Questions:
- why does Proposition 8 imply that $z_\infty$ is nondegenerate? Why, given its definition 1 line above, is it of the form written in (PrC)?
- In prop 9 shouldn't the limit of $\epsilon_n$ be $\infty$?


Minor:
- In theorem 3, $\hat A$ is such that $\hat \pi = \text{Sink}(\hat A, \epsilon$; this has been introduced earlier but given the large number of variables defined in the paper, it may not hurt to recall it here; in addition, before, Sink was applied to $c$, so it should be $\text{Sink}(c_{\hat A}, \epsilon$. In prop 10, prop 11 the order of the arguments is reversed.
- can the authors explain the name "precertificate" (what's "pre" about it? Is there a difference with what Dunner and coauthors call "Dual certificate" in https://arxiv.org/abs/1602.05205)?
- The authors refer to "condition PrC", but that is just an equality. Condition PrC would be that the dual norm of the precertificate outside the support is $< \lambda$
- what's "the convex dual of W (A)"? convex conjugate of a function (as in Proposition 4)/convex dual of a convex optimization problem?
- is the notation $\langle c, A \rangle$ for a non scalar result?
- In theorem 7 I spent quite some time looking for what the number $m$ was in other parts of the paper (because of the analogy $m/n$ in compressed sensing) before realizing that it was a free paramert; maybe replacing it by $\ln(1/\delta)$ is more common (that is only a suggestion).
- below 9, in $A_n$ definition, both $\epsilon$'s should be $\epsilon_n$? Same in Prop 11
- it follows (see e.g. Hiriart-Urruty et al. (1993)) : can you point to a specific result in the book? Also in the bibliography, the authors names appear twice for this book.

Typos:
- the paper does not seem to use the unmodified iclr template, the font differs from that of other papers
- in Problem iOT L1 hat Pi, the first argument of L should be $A$ not $c_A$ (see def of L 2 equations above)
-  The iOT problem of iOT
- in a series of paper*s*
- Thm 3: the solution ... satisf*ies*
-  minimial norm
- sufficinetly$
- result above implies that for provided that the number
- as exposed in Section ??
- Cauchy Schwartz
- see Proposition PrC (it's an equation not a proposition, and I don't see why pRC shows that W is $C^2$)

---

> ### Author Response · Authors · 2023-11-14
>
> Thank you for your comments.
>
> > The authors could really afford to improve the pedagogy of the paper, which is quite heavy in terms of notation. Theory is quite involved, require background references to other works such as Carlier or Galichon. The experiment description is a big block which, in my opinion, does not bring as many insights as it could.
>
> We have added an ``intuitions'' section to explain the relation between the certificate and optimality conditions. There is also now an "interpretation" after Propositions 8 and 9. Regarding the experiments, we added an additional example for the anti-symmetric case and clarified the interpretation of the results.
>
> > Why does Proposition 8 imply $z_\infty$ is non-degenerate? Why, given its definition 1 line above, is it of the form written in (PrC)?
>
> $z_\infty$ is actually the dual solution for $\lambda$ fixed. It corresponds, in the original submission, to $z^\lambda$ of Proposition 5. We don't claim that $z_\infty$ has the form  written in (PrC) in the original solution. What we have is that $\lim_{\lambda \to 0} z_\infty\to z_A^\ast$ and hence, non-degeneracy of $z_A^\ast$ implies that the magnitude of $z_\infty$ outside the support of $A$ is less than one (for $\lambda$ sufficiently small).
>
> We agree with the reviewer that this was not  clear in the original submission and we have changed this in the new submission it to make it so.
>
> > In prop 9 shouldn't the limit of $\epsilon_n$	 be $\infty$ ?
>
> There was in fact a typo and in the $\epsilon\to \infty$ proposition in the Gaussian case. We have corrected this.
>
> > In theorem 3, $\hat A$
> is such that $\text{Sink}(\hat{A},\epsilon)$ ; this has been introduced earlier but given the large number of variables defined in the paper, it may not hurt to  recall it here; in addition, before, Sink was applied to $c$, so it should be $\text{Sink}(c_{\hat{A}},\epsilon$. In prop 10, prop 11 the order of the arguments is reversed.
>
> We agree with the reviewer and we have modified the new submission accordingly.
>
> > Can the authors explain the name ``precertificate'' (what's ``pre'' about it? Is there a difference with what Dunner and coauthors call  ``Dual certificate'' in https://arxiv.org/abs/1602.05205)?
>
> The use of ``pre'' was to make a distinction with the minimal norm certificate defined afterwards. Under non-degeneracy the two are the same but not necessarily in the degenerate case. As this distinction was hurting readability, we have changed the name and reformulated the section. Moreover, we moved to the appendix the connection with duality.
>
> > The authors refer to ``condition PrC'', but that is just an equality. Condition PrC would be that the dual norm of the precertificate outside the support is $<\lambda$
>
> We agree with the reviewer and this now reflected in the reformulated section.
>
> >  what's ``the convex dual of $W (A)$''? convex conjugate of a function (as in Proposition 4)/convex dual of a convex optimization problem?
>
> We agree with the reviewer that this was not the best terminology and we modified it.
>
> >  is the notation $\langle c,A\rangle$ for a non scalar result?
>
> The notation $\langle c, A\rangle$ corresponds to a function: given a set of fixed cost functions $\lbrace C_1(\cdot,\cdot),\ldots, C_t(\cdot,\cdot)\rbrace$, the notation meant a linear combination of those with coefficients given by the vector $A$, \emph{i.e.,}
> \begin{align*}
>     \langle c, A\rangle(x,y):= \sum_j A_j C_j(x,y)
> \end{align*}
> In fact, $\langle c,A\rangle(\cdot,\cdot)$ was just another notation for $c_A(\cdot,\cdot)$ that was meant to highlight the linear dependence of $c_A(\cdot,\cdot)$  on $A$. Given that it was never re-used, we decided to drop it to avoid confusion with the inner product notation.
>
> > In theorem 7 I spent quite some time looking for what the number $m$ was in other parts of the paper (because of the analogy $m/n$ in compressed sensing) before realizing that it was a free paramert; maybe replacing it by $\log(1/\delta)$ is more common (that is only a suggestion).
>
> We welcome the suggestion which was integrated in the new submission.
>
> > below 9, in $A_n$ definition, both $\epsilon$'s should be $\epsilon_n$? Same in Prop 11
>
> Yes. This was a typo that is now corrected.
>
> > it follows (see e.g. Hiriart-Urruty et al. (1993)) : can you point to a specific result in the book? Also in the bibliography, the authors names appear twice for this book.
>
> the specific result is now mentioned and the bibliography was corrected.
>
>
> > Typos
>
> All the typos suggested by the reviewer were corrected.

---

### Meta-Review · Area_Chair_bSvK · 2023-12-06

**Metareview:**

This paper is a theoretical study of the l1 penalty for the inverse optimal transport problem. The authors show that iOT is related to graphical lasso and classical lasso. Some reviewers viewed the contribution as above the bar because it provides a more solid theoretical grounding for the iOT method. It addresses an open problem in the field. However, there were concerns that the paper was less accessible to the community and is more targeted towards those already quite familiar with iOT. The work is theoretical lacks computational timing and experiments with real data sets. Overall, the paper is an interesting theoretical contribution and may serve as a foundation for further theoretical and experimental work.

**Justification For Why Not Higher Score:**

The two reviews that advocated for acceptance where quite brief and lacked detail.

**Justification For Why Not Lower Score:**

The overall sentiment of the ratings is towards acceptance.

---

### Decision · Program_Chairs · 2024-01-16

Accept (poster)